# SETD3 protein is the actin-specific histidine *N*-methyltransferase

Sebastian Kwiatkowski[1†], Agnieszka K Seliga[1†], Didier Vertommen[2], Marianna Terreri[1], Takao Ishikawa[3], Iwona Grabowska[4], Marcel Tiebe[5,6], Aurelio A Teleman[5,6], Adam K Jagielski[1], Maria Veiga-da-Cunha[7]*, Jakub Drozak[1]*

[1]Department of Metabolic Regulation, Faculty of Biology, University of Warsaw, Warsaw, Poland; [2]Protein Phosphorylation Unit, de Duve Institute, Université Catholique de Louvain, Brussels, Belgium; [3]Department of Molecular Biology, Faculty of Biology, University of Warsaw, Warsaw, Poland; [4]Department of Cytology, Faculty of Biology, University of Warsaw, Warsaw, Poland; [5]German Cancer Research Center (DKFZ), Heidelberg, Germany; [6]Heidelberg University, Heidelberg, Germany; [7]Metabolic Research Unit, de Duve Institute, Université Catholique de Louvain, Brussels, Belgium

*For correspondence:
maria.veiga@uclouvain.be (MV–C);
jdrozak@biol.uw.edu.pl (JD)

[†]These authors contributed equally to this work

Competing interests: The authors declare that no competing interests exist.

**Abstract** Protein histidine methylation is a rare post-translational modification of unknown biochemical importance. In vertebrates, only a few methylhistidine-containing proteins have been reported, including β-actin as an essential example. The evolutionary conserved methylation of β-actin H73 is catalyzed by an as yet unknown histidine *N*-methyltransferase. We report here that the protein SETD3 is the actin-specific histidine *N*-methyltransferase. In vitro, recombinant rat and human SETD3 methylated β-actin at H73. Knocking-out SETD3 in both human HAP1 cells and in *Drosophila melanogaster* resulted in the absence of methylation at β-actin H73 in vivo, whereas β-actin from wildtype cells or flies was > 90% methylated. As a consequence, we show that Setd3-deficient HAP1 cells have less cellular F-actin and an increased glycolytic phenotype. In conclusion, by identifying SETD3 as the actin-specific histidine *N*-methyltransferase, our work pioneers new research into the possible role of this modification in health and disease and questions the substrate specificity of SET-domain-containing enzymes.
DOI: https://doi.org/10.7554/eLife.37921.001

## Introduction

Protein methylation is one of the most common post-translation modifications (PTMs) in eukaryotic cells (for a review, see *Clarke, 2013*) and histones are the best studied substrates of protein *N*-methyltransferases. Consequently, much is known about the methylation of lysine and arginine residues in histones and about the fundamental role of such modifications in the epigenetic control of mammalian gene expression (*Greer and Shi, 2012*). In addition, there are a growing number of examples showing that the methylation of non-histone proteins is a prevalent PTM that regulates diverse biological processes, including protein synthesis and signal transduction (for a review, see *Biggar and Li, 2015*). More interestingly, non-histone proteins may be methylated on atypical residues such as glutamate, glutamine, cysteine and histidine (*Clarke, 2013*). The methylation of histidine residues in proteins has been known for many years but, curiously, only for a few proteins in Nature, including β-actin (*Johnson et al., 1967*), S100A9 protein (*Raftery et al., 1996*), myosin (*Elzinga and Collins, 1977*), myosin kinase (*Meyer and Mayr, 1987*) and ribosomal protein RPL3 (*Webb et al., 2010*). However, a recent study from *Ning et al. (2016)* suggests that protein methylation on histidine residues may actually be a quite common phenomenon, modifying dozens if not hundreds of intracellular proteins in mammalian cells.

The current knowledge on histidine *N*-methyltransferases, which are the enzymes that catalyze the methylation of histidines in proteins is still sparse. The yeast protein Yil110w (Hpm1p), which methylates histidine 243 in the ribosomal protein Rpl3p, is the first and only histidine protein methyltransferase molecularly identified to date (*Webb et al., 2010*). Hpm1p is the seven β-strand methyltransferase (7BS MTase), which has been classified in the methyltransferase family 16 (MTF16) (for review, see *Falnes et al., 2016*). The MTF16 group comprises mostly protein-lysine methyltransferases, such as Mettl20–23 and Camkmt, with the noticeable exceptions of Hpm1p and of Efm7, which methylates the N-terminal glycine in the eukaryotic translation elongation factor one alpha (eEF1A). The MTF16 group appears to be distinct from both the other known 7BS protein-lysine MTases (e.g. Dot1L, Mettl10 and N6amt2) and the non-protein methylating 7BS MTases (e.g. Mettl1, Comt and Gnmt). On the other hand, the enzyme responsible for actin-specific histidine methylation has been partially purified, but the gene encoding this enzyme remains to be identified (*Vijayasarathy and Narasinga Rao, 1987*; *Raghavan et al., 1992*).

Actin is one of the most abundant proteins in eukaryotic cells and a major component of the cytoskeleton (for a review, see *Pollard and Cooper, 2009*). It is a highly conserved protein during evolution (≈90% identity between yeast and human protein), and in vertebrates there are three main isoforms that only differ by a few amino acids at their N-terminus. α-Actins are expressed in skeletal, cardiac and smooth muscle, whereas β- and γ-isoforms are present in non-muscle and muscle cells. Under physiological conditions, actin exists as a 42-kDa monomeric globular protein (G-actin) that binds ATP and spontaneously polymerizes into stable filaments (F-actin). Once these filaments have been assembled, the terminal phosphate of the bound ATP is hydrolyzed and slowly released from actin, leading to the sequential depolymerization of the filament. The maintenance of a pool of actin monomers, the initialization of the polymerization process, and the regulation of the assembly and turnover of actin filaments are extremely complex and well controlled. These involve more than 100 interacting partner proteins, so that actin contributes to more protein–protein interactions than any other protein (*Lappalainen, 2016*). Actin plays a role in many central cell functions, including cell growth, division, and motility (*Dominguez and Holmes, 2011*), which makes it essential for cell survival. Consequently, various efforts have been made to elucidate the role of the conserved methylation of the H73 residue (*Nyman et al., 2002*). As a result, it was shown in vitro that H73 methylation stabilizes actin filaments, and that the rate of filaments depolymerization was increased when actin was non-methylated (*Nyman et al., 2002*). Nevertheless, the physiological effects of actin hypomethylation have yet to be reported in cells, tissues or organisms.

Here, we report the identification of actin-specific histidine *N*-methyltransferase (EC 2.1.1.85) as SET domain-containing protein 3 (SETD3, C14orf154), which was previously shown to act as a histone lysine methyltransferase (*Eom et al., 2011*). We reveal that SETD3 is responsible for the methylation of H73 in human and drosophila actin, and that a loss of SETD3 activity impacts the phenotype of the human HAP1 cell line by lowering F-actin content and increasing glycolytic activity.

## Results

### Purification and identification of rat actin-specific histidine *N*-methyltransferase

Actin-specific histidine *N*-methyltransferase activity was assayed by measuring the incorporation of the [³H]methyl group of [³H]*S*-adenosyl methionine ([³H]SAM) into either homogeneous recombinant human β-actin produced in *Escherichia coli* or a H73A β-actin mutant (*Figure 1—figure supplement 1*), which served as a negative control, ensuring that only the H73-specific *N*-methyltransferase activity was measured (*Figure 1—figure supplement 2*). The use of recombinant human β-actin as the rat methyltransferase substrate was reasonable because the sequences of human and rat β-actin cannot be distinguished from one another, owing to an extreme evolutionary conservation. Following numerous purification steps, the *N*-methyltransferase activity was purified from rat leg muscle approximately 1200-fold (see *Table 1*). The methyltransferase was eluted as a single peak in each of the purification steps (*Figure 1*), indicating the presence of a single enzyme species. No methylation of H73A β-actin was detected at any stage of the purification, suggesting that the myofibrillar extract of rat muscle is devoid of other β-actin-methylating enzymes. The overall yield of the

**Table 1.** Purification of actin-specific histidine *N*-methyltransferase from rat skeletal muscles

| Fraction | Volume | Total protein | Total activity | Specific activity | Purification | Yield |
|---|---|---|---|---|---|---|
| | *ml* | *mg* | *pmol min$^{-1}$* | *pmol min$^{-1}$ mg$^{-1}$* | *-fold* | *%* |
| Myofibrillar extract (20,000 × *g* supernatant) | 2400 | 35,273 | 8768 | 0.248 | 1 | 100 |
| 20% PEG fraction | 1000 | 16,333 | 4853 | 0.297 | 1.2 | 55.4 |
| DEAE Sepharose | 84.5 | 2757 | 5753 | 2.09 | 8.4 | 65.7 |
| Q Sepharose | 63.7 | 800 | 4507 | 5.637 | 22.7 | 51.5 |
| Phenyl Sepharose | 93.4 | 66 | 964 | 14.602 | 58.8 | 11.0 |
| HiScreen Blue | 27 | 14.3 | 682 | 47.6 | 191.7 | 7.8 |
| Superdex 200[#] | 3 | 1.20 | 185.5 | 154.8 | 623.4 | 2.12 |
| Reactive Red 120[#] | 3 | 0.18 | 52.04 | 291.2 | 1172.5 | 0.59 |

[#] The data represent mean values for the most purified fraction of the step applied. PEG, polyethylene glycol.

DOI: https://doi.org/10.7554/eLife.37921.006

purification was only about 0.6% (see *Table 1*), largely because of the fact that in the last two steps of purification, only the most active fractions were used for the next step of the procedure.

SDS-PAGE analysis of the activity peak fractions (F2 and F3) from the Reactive Red 120-agarose showed a protein concentration that was too low to allow the identification of the proteins coeluting with the enzymatic activity (*Figure 1D*). Thus, the proteins in these fractions were concentrated about 15-fold and reanalyzed by SDS-PAGE. Actin-methylating activity coeluted with three protein bands of about 65, 75 and 90 kDa, which were clearly visible in the concentrated fraction F3 (*Figure 1D*). These bands were cut out from the gel, digested with trypsin and analyzed by MS/MS. The sequences of the identified peptides were compared with the rat reference proteome from the NCBI Protein database. This comparison indicated that there was only one methyltransferase among these peptides, present in band 'C', and this was identified as histone lysine *N*-methyltransferase SETD3 (*Table 2*). Eighteen peptide sequences from the rat SETD3 (underlined in *Figure 2*) that had shared identity with other SETD3 proteins in the National Center for Biotechnology Information (NCBI) Protein database were found to cover about 39% of its sequence. However, to exclude the possibility of missing any potential methyltransferase as the result of a poor extraction of tryptic peptides from protein bands, we performed the MS/MS identification of all proteins present in the entire fraction 14 of Superdex 200 (see *Figure 1B*) and fraction F3 of Reactive red 120 (see *Figure 1C*), and SETD3 protein was again the only methyltransferase found in these fractions.

As SETD3 protein was shown to methylate histones (*Eom et al., 2011*) and all of the SET-domain-containing enzymes characterized to date are protein lysine methyltransferases (*Clarke, 2013*), we were surprised to find that SETD3 was the protein histidine *N*-methyltransferase responsible for the measured activity. To double check our results, we performed another complete, though modified, round of purification of the actin-methylating enzyme from rat leg muscles, as well as MS/MS analysis of the most purified fractions from Superdex 200. Once again, we identified SETD3 protein as the only meaningful candidate for rat actin-specific histidine *N*-methyltransferase, suggesting we had identified the correct protein. Moreover, our results were in agreement with previous reports suggesting that the actin-specific histidine *N*-methyltransferase corresponded to a ≈70 kDa protein (*Raghavan et al., 1992*).

## Analysis of SETD3 protein sequences

Protein BLAST searches (*Altschul et al., 1997*) with rat SETD3 and phylogenetic analysis of the resulting sequences indicated that orthologs of this protein were found in almost all eukaryotes including vertebrates (≈75–98% identity with the rat sequence), insects (≈35–40% identity), plants and fungi (≈30% identity) (*Figure 3*). No SETD3 orthologs were detected in the sequenced proteome of *Saccharomyces cerevisiae* or of *Naegleria gruberi* in agreement with previous reports, showing the absence of actin-specific histidine methyltransferase activity in these species (*Kalhor et al., 1999*). All taxa followed the expected lines of descent, indicating that the enzyme was present in a common eukaryotic ancestor (see *Figure 3*). A low similarity in amino-acid sequence between

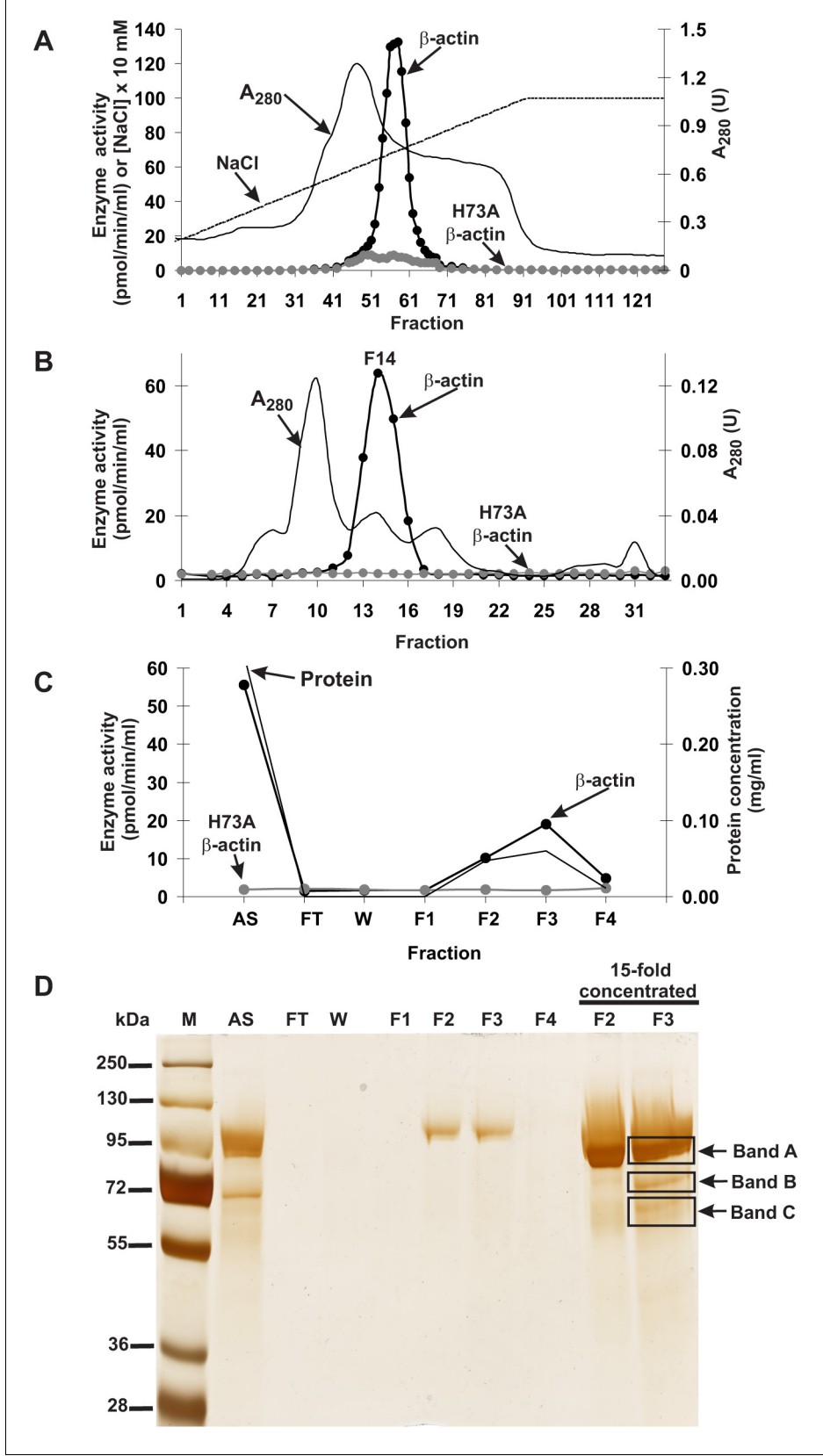

**Figure 1.** Purification of the rat actin-specific histidine *N*-methyltransferase. The enzyme was purified by chromatography on (**A**) DEAE-Sepharose, (**B**) Q-Sepharose, Phenyl-Sepharose, HiScreen Blue-Sepharose (shown in *Figure 1 continued on next page*

*Figure 1 continued*

*Figure 1—figure supplement 3*), Superdex 200, and (C) Reactive Red 120 Agarose, as described in the 'Materials and methods' section. Fractions were tested for actin-specific histidine *N*-methyltransferase activity using either homogeneous recombinant human β-actin or its mutated form H73A. (D) The indicated fractions eluted from the Reactive Red 120 Agarose column were analyzed by SDS-PAGE and the gel was silver-stained (*Shevchenko et al., 1996*). M, prestained protein marker; AS, applied sample (Fraction 14 from Superdex 200); FT, flow through; W, wash. Fractions F1, F2, F3 and F4 were eluted with 200, 400, 900 and 1400 mM NaCl, respectively. In addition, fractions F2 and F3 were concentrated 15-fold by ultrafiltration (Vivaspin 500) before being analyzed. The indicated bands were cut out of the gel for mass spectrometry analysis.

DOI: https://doi.org/10.7554/eLife.37921.002

The following figure supplements are available for figure 1:

**Figure supplement 1.** (A) SDS-PAGE and (B) Western-blot analysis of fractions obtained during the purification of recombinant human β-actin overexpressed in *E. coli*.

DOI: https://doi.org/10.7554/eLife.37921.003

**Figure supplement 2.** The *N*τ-methylation of the actin H73 residue catalyzed by mammalian actin-specific histidine *N*-methyltransferase (EC 2.1.1.85).

DOI: https://doi.org/10.7554/eLife.37921.004

**Figure supplement 3.** Purification of the rat actin-specific histidine *N*-methyltransferase.

DOI: https://doi.org/10.7554/eLife.37921.005

eukaryotic SETD3 orthologs and some bacterial proteins was also detected (e.g. ≈30% identity, *Cystobacter fuscus,* NCBI Reference Sequence: WP_095987699.1), suggesting that the eukaryotic enzyme might have been acquired from an ancestral prokaryote.

All identified SETD3 proteins contain the SET and Rubisco LSMT substrate-binding domains at their N- and C-terminus, respectively. The SET domain is thought to interact directly with the protein substrate and most probably contributes indirectly to SAM binding too, whereas the Rubisco

**Table 2.** Proteins identified in the gel bands submitted to trypsin digestion and MS/MS analysis. Identified proteins are listed for each band according to their score as calculated using ProteinLynx Global Server software (PLGS).
For each protein, the molecular weight (mW) and sequence coverage are also indicated. Occasional peptide hits corresponding to keratins have not been included.

| Gel band | Protein name | NCBI Protein accession number | PLGS score[*] | mW (Da) | Coverage (%) |
|---|---|---|---|---|---|
| A | Elongation factor 2 | NP_058941.1 | 16,249 | 95,222 | 73.5 |
| B | Calpain one catalytic subunit | NP_062025.1 | 3024 | 82,067 | 47.7 |
| | Elongation factor 2 | NP_058941.1 | 789 | 95,222 | 15.2 |
| | Nuclear protein localization protein four homolog | NP_542144.1 | 778 | 68,013 | 31.7 |
| C | Elongation factor 2 | NP_058941.1 | 2024 | 95,222 | 42.1 |
| | Nuclear protein localization protein four homolog | NP_542144.1 | 1263 | 68,013 | 37.3 |
| | Histone lysine *N*-methyltransferase SETD3 | XP_002726820.2 | 1136 | 67,378 | 39.1 |
| | Eukaryotic peptide chain release factor GTP binding subunit ERF3A | NP_001003978.1 | 505 | 68,708 | 41.2 |
| | Guanylate binding protein 1 | NP_598308.1 | 404 | 67,312 | 21.8 |
| | Filamin-B-like isoform X2 | XP_003751493.2 | 272 | 121,901 | 1.4 |

[*]PLGS Score is calculated by the ProteinLynx Global Server (v2.4) software using a Monte Carlo algorithm to analyze all acquired mass spectral data and it is a statistical measure of accuracy of assignation. A higher score implies a greater confidence of protein identity.

DOI: https://doi.org/10.7554/eLife.37921.008

The following source data is available for Table 2:

Source data 1. Mass spectrometry (Q-TOF) identification of proteins present in the gel bands submitted to trypsin digestion.

DOI: https://doi.org/10.7554/eLife.37921.009

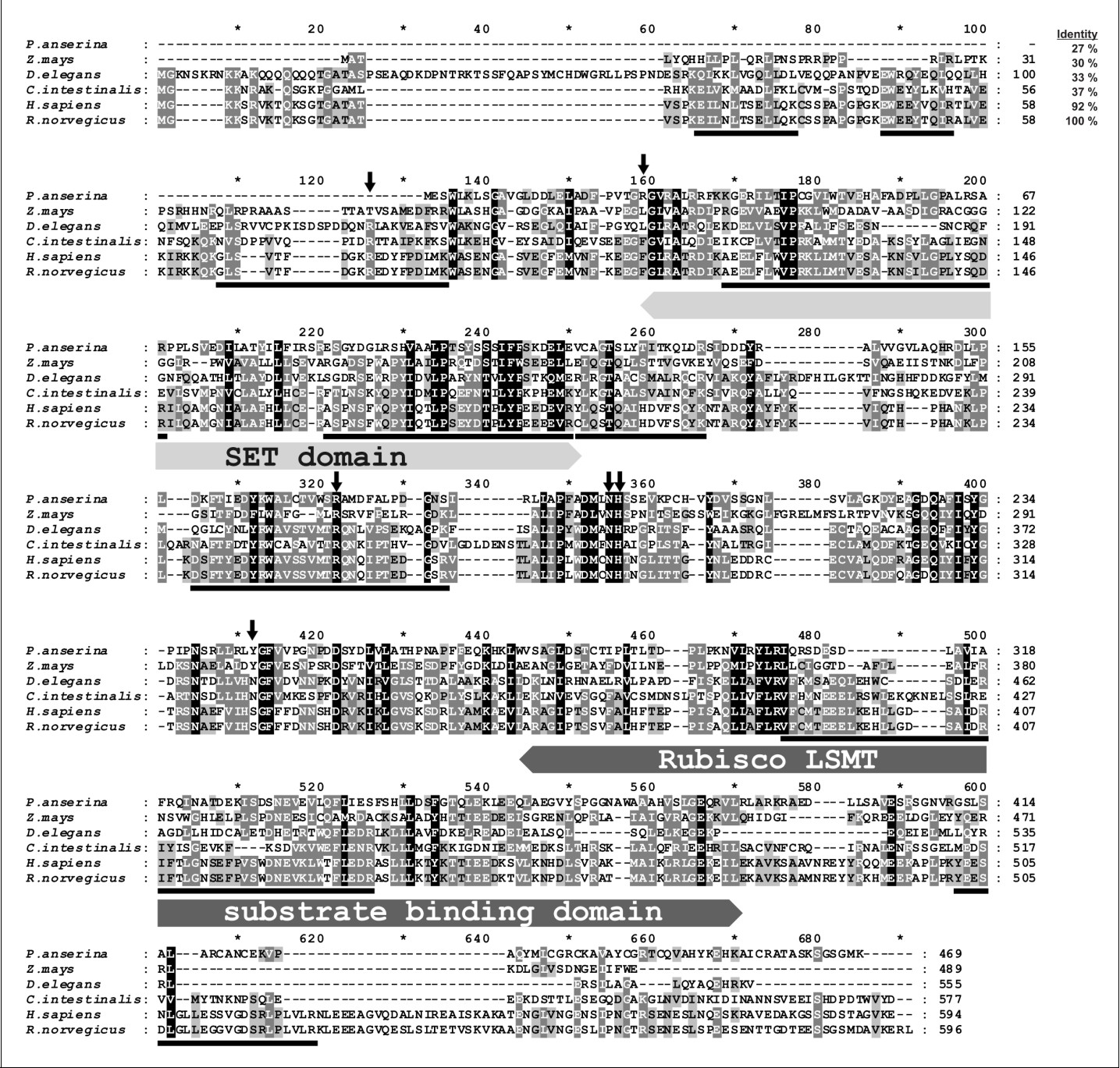

**Figure 2.** Amino-acid sequence alignment of the rat SETD3 protein with its orthologs. Sequences of rat (*Rattus norvegicus*, XP_002726820.2), human (*Homo sapiens*, NP_115609.2), *Ciona intestinalis* (*C. intestinalis*, XP_002131202.1), *Drosophila elegans* (*D. elegans*, XP_017114801.1), *Podospora anserina* (*P. anserina*, CDP29262.1) and *Zea mays* (*Z. mays*, NP_001168589.1) protein were obtained from the National Center for Biotechnology Information (NCBI) Protein database. Both the rat and the human sequences have been confirmed by PCR amplification of the cDNA and DNA sequencing. The percentage of amino-acid identities with the rat SETD3 protein is given in the top right corner of the figure. The conserved protein substrate-binding domains (SET and Rubisco large subunit methyltransferase (LSMT) substrate binding) are labeled above the alignment, while amino-acid residues that interact with *S*-adenosyl-L-methionine (SAM) are indicated by arrows, as inferred from the crystal structure of the human SETD3 enzyme (PDB 3SMT). The peptides identified by mass spectrometry in the protein purified from rat leg muscle are underlined in the rat sequence. The level of residue conservation is indicated by black (100%), dark grey (70% and more) and light gray (50% and more) background.
DOI: https://doi.org/10.7554/eLife.37921.007

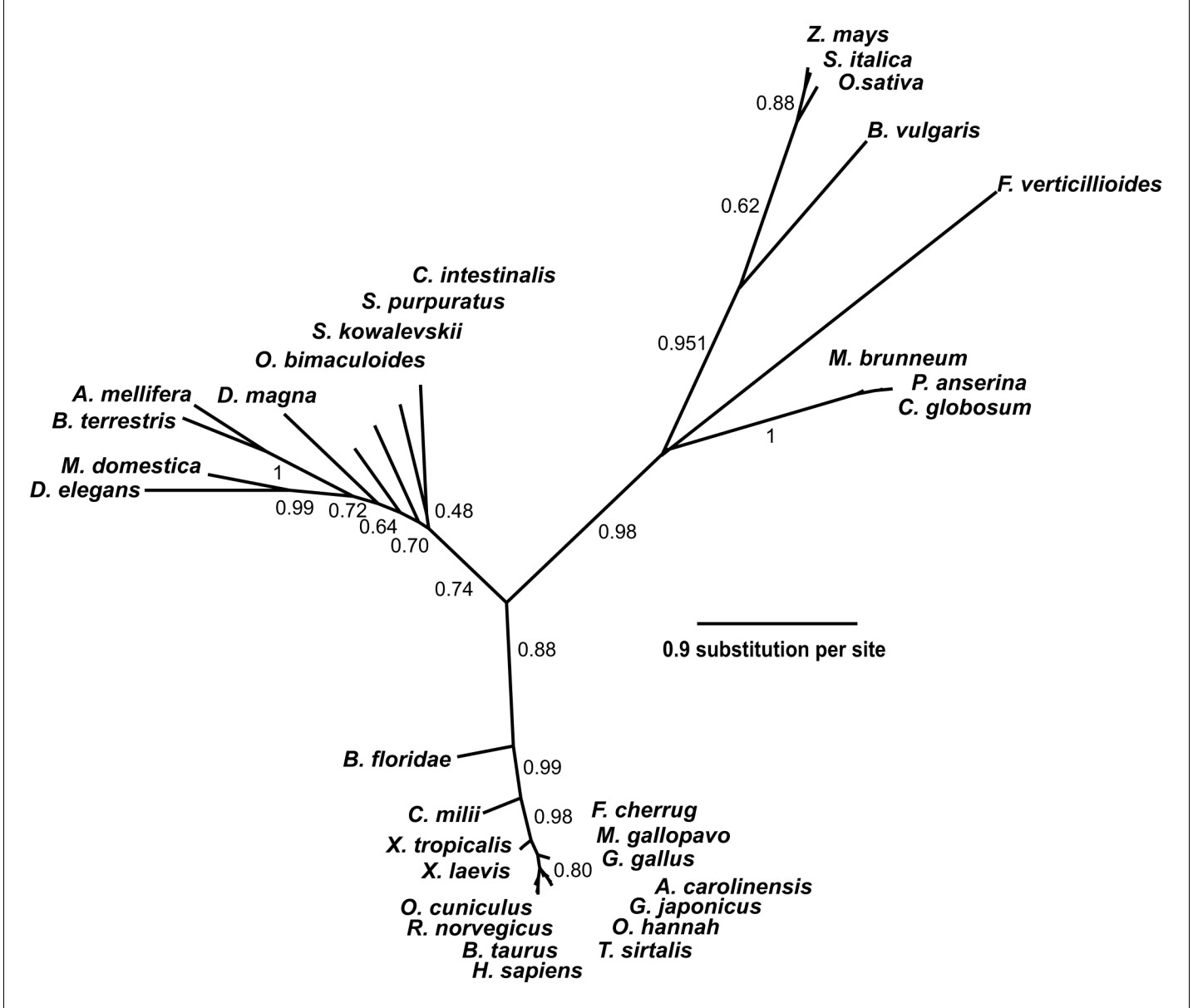

**Figure 3.** Phylogenetic tree of the SETD3 proteins. Protein sequences were aligned using Muscle (*Edgar, 2004*), and the phylogenetic tree was inferred with the use of PhyML (*Guindon and Gascuel, 2003*) implemented in phylogeny.fr web service (*Dereeper et al., 2008*). Branch support values assessed using the aLRT test are indicated (*Anisimova and Gascuel, 2006*). The protein sequences used for the analysis are as follows: *Anolis carolinensis* (XP_003214383.2); *Apis mellifera* (XP_016770011.1); *Beta vulgaris* (XP_010683122.1); *Bombus terrestris* (XP_003398458.1); *Bos taurus* (XP_589822.3); *Branchiostoma floridae* (XP_002596839.1); *Callorhinchus milii* (XP_007906724.1); *Chaetomium globosum* (XP_001224775.1); *Ciona intestinalis* (XP_002131202.1); *Daphnia magna* (KZS12928.1); *Drosophila elegans* (XP_017114801.1); *Falco cherrug* (XP_005438913.1); *Fusarium verticillioides* (XP_018752240.1); *Gallus gallus* (NP_001006486.1); *Gekko japonicus* (XP_015275964.1); *Homo sapiens* (NP_115609.2); *Meleagris gallopavo* (XP_003206761.2); *Metarhizium brunneum* (XP_014542924.1); *Musca domestica* (XP_011294563.1); *Octopus bimaculoides* (XP_014787293.1); *Ophiophagus hannah* (ETE71402.1); *Oryctolagus cuniculus* (XP_008247172.2); *Oryza sativa* (XP_015651332.1); *Podospora anserina* (CDP29262.1); *Rattus norvegicus* (XP_002726820.2); *Saccoglossus kowalevskii* (XP_006819296.1); *Setaria italica* (XP_004956796.1); *Strongylocentrotus purpuratus* (XP_798530.2); *Thamnophis sirtalis* (XP_013914404.1); *Xenopus laevis* (OCT68299.1); *Xenopus tropicalis* (XP_012823880.1); and *Zea mays* (NP_001168589.1).
DOI: https://doi.org/10.7554/eLife.37921.010

LSMT substrate-binding domain seems to interact with a protein substrate only (see *Figures 2* and *4*). Although the sequence identity between orthologs from distinct species such as rat and maize (*Zea mays*) was rather low (about 30%), the alignment of their sequences showed the presence of

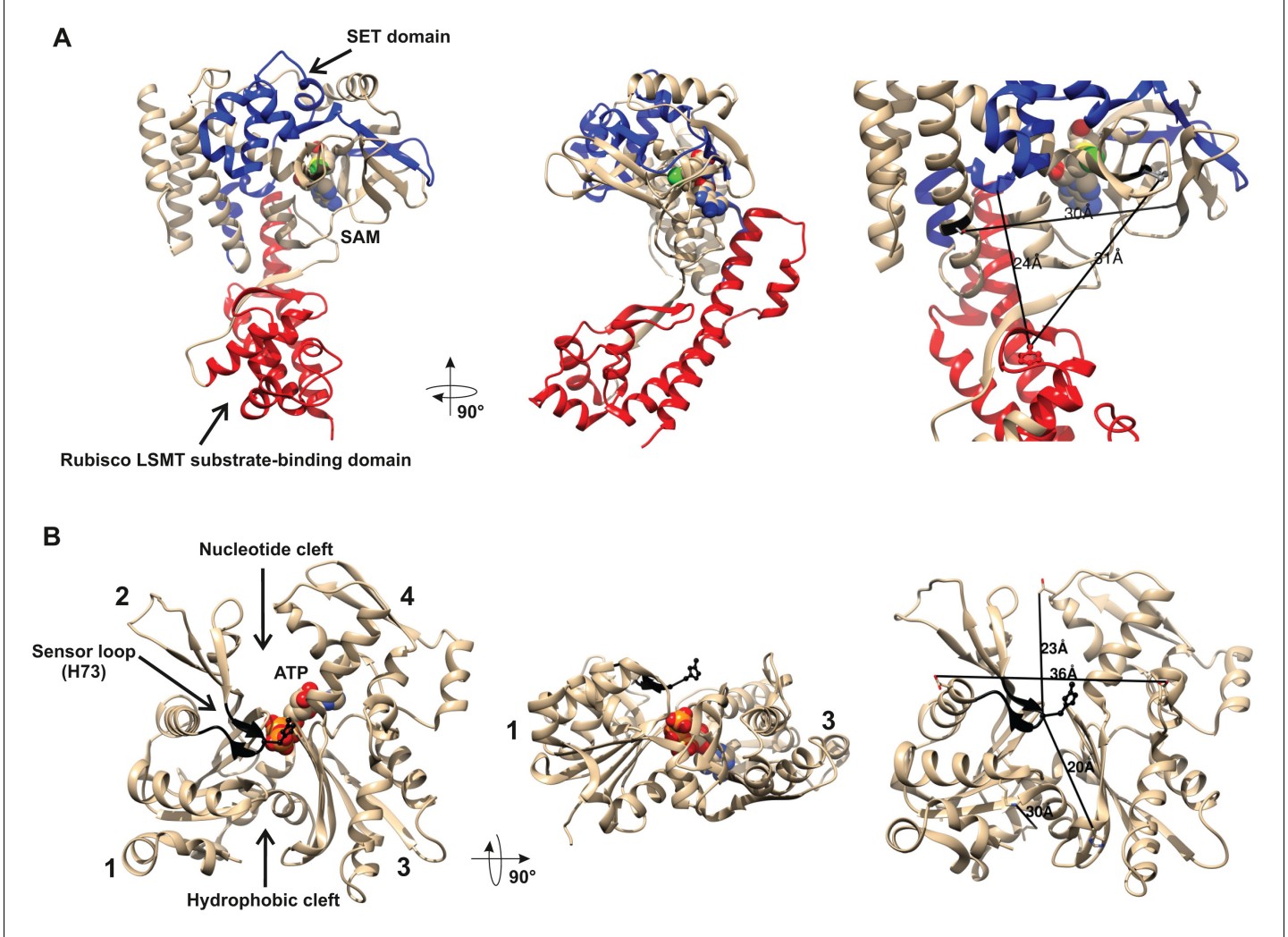

**Figure 4.** Structures of human β-actin and SETD3 enzyme. Ribbon representations of the structures of both (**A**) the SETD3 protein and (**B**) the actin monomer are shown in different projections. (**A**) The 'SET' and 'Rubisco LSMT' substrate-binding domains of SETD3 are highlighted in blue and red, respectively. The spatial organization of these domains gives the enzyme a cradle-like shape. The *S*-adenosyl-L-methionine (SAM) molecule is bound at the bottom of this cradle, exposing its labile methyl group (green) to a protein substrate. (**B**) The four subdomains of actin are marked with numbers. Subdomains 1 and 2 form the small (outer) domain, whereas subdomains 3 and 4 constitute the large (inner) domain. Two large clefts are present between these domains: the nucleotide binding cleft and the hydrophobic one, while ATP is bound at the center of the molecule. The nucleotide sensor loop, which contains the methyl-accepting H73 residue, spans from Pro70 to Asn78 and is shown in black. Several residue–residue distances are shown to give information concerning the molecular dimension of these proteins. All models were prepared using UCSF Chimera (*Pettersen et al., 2004*) from the Protein Data Bank (PDB) structures of β-actin (2BTF) and SETD3 (3SMT).

DOI: https://doi.org/10.7554/eLife.37921.011

several motifs that were well conserved in both domains, suggesting that these orthologs may have similar substrates (see *Figure 2*).

A partial protein structure of human SETD3 has been published (PDB 3SMT). It lacks about 100 amino acids at the C-terminus, but it shows both substrate-binding domains. As shown in *Figure 4A*, these two domains form a wide cleft in the center of SETD3, so that the enzyme adopts a cradle-like shape. The SAM molecule is bound in the bottom of the cleft, exposing its labile methyl group (see *Figure 4A*). It seems likely that the cradle-shape of the enzyme allows it to arch over a protein substrate, and that the position of SAM facilitates the subsequent transfer of the methyl group onto a specific amino acid (in the case of actin, a histidine), most probably located at the surface of a substrate protein.

As shown in *Figure 4B*, β-actin has a globular shape, flattened in one dimension. It consists of large and small domains that are further subdivided into subdomains 1, 2 and 3, 4, respectively. The nucleotide-binding cleft (for ATP or ADP binding) is located inside the cleft between subdomains 2 and 4, while a hydrophobic cleft is present between subdomains 1 and 3 at the 'bottom' of the actin molecule, and serves as a primary binding site for numerous actin-binding proteins. The methyl-accepting H73 residue is present in a nucleotide sensor loop (P70 to N78) in an insert between subdomains 1 and 2. The H73 is present at the surface of the actin monomers and seems to be easily accessed by the methyltransferase. Furthermore, the estimated dimensions of the actin monomer and of the substrate-binding cleft in SETD3 indicate that actin could be accommodated well at the cleft of the methyltransferase and that steric hindrance should not prevent interactions between these proteins.

## Characterization of recombinant SETD3 proteins

To confirm the molecular identity of rat actin-specific histidine *N*-methyltransferase with that of rat SETD3 and to compare the activities of rat and human methyltransferases, these two SETD3 orthologs were expressed in COS-7 cells as fusion proteins with the C-terminal polyhistidine tag (*Figure 5*). The recombinant enzymes catalyzed the methylation of recombinant β-actin, but not of the H73A mutant (see *Figure 5*). In further studies, large amounts of recombinant enzymes were produced in COS-7 cells, purified to homogeneity (*Figure 6—figure supplement 1*) and shown to catalyze the methylation of actin (*Figure 6*). In addition, the recombinant human and rat SETD3 with the N-terminal His$_6$-tag were produced in *E. coli*, purified (*Figure 7—figure supplement 1*) and also shown to methylate actin (*Figure 7*). Their specific activity was similar to that determined for the enzymes produced in the COS-7 cells ($\approx$5 nmol.min$^{-1}$.mg$^{-1}$ protein), which further confirms the identification of SETD3 as the actin-specific methyltransferase.

As partially purified rabbit actin-specific histidine *N*-methyltransferase (*Raghavan et al., 1992*) was shown to methylate a synthetic peptide (Peptide H: YPIE**H**GIVT), corresponding to the H73-containing nucleotide sensor loop of the actin molecule, we wanted to know whether SETD3 can methylate peptide H. As shown in *Figure 8*, the two SETD3 orthologs catalyzed the methylation of peptide H, but they did not methylate peptide A (YPIE**A**GIVT), a variant of the nucleotide sensor loop in which H73 is replaced by an alanine. Even if the rat and human SETD3 activities measured with peptide H were very poor ($\approx$ 0.02 nmol.min$^{-1}$.mg$^{-1}$ protein) compared with the actual methylation of actin ($\approx$ 5 nmol. min$^{-1}$.mg$^{-1}$ protein), these activities further indicated that SETD3 is the histidine-specific *N*-methyltransferase.

To confirm the specific methylation of H73 in recombinant human β-actin, we incubated it with homogenous recombinant rat and human SETD3 enzymes in the presence of deuterated SAM ([$^2$H] SAM). This resulted in approximately 35% of the actin being methylated, as measured using a radiochemical assay that was run in parallel. At the end of the reaction, the proteins in the assay mixture were separated by SDS-PAGE, and the bands corresponding to the β-actin polypeptide were analyzed by nanoUPLC-Q-TOF. At least 74% of the actin sequence coverage was achieved in each MS analysis, including all of the known methyl-accepting sites in human β-actin (K18, K68, H73, K84, K191 and K326, see http://www.phosphosite.org). Peptide YPIEHGIVTNWDDMEK (M + H = 1963.9355 Da) containing the H73 residue was the only tri-deuterium-methylated peptide detected by the PLGS 2.4 software (Waters, USA). Manual inspection of the MS/MS spectra of both the parent ion of this peptide and its fragments confirmed the presence of tri-deuterium-methylated H73 (*Figure 9*). Both rat and human SETD3 enzymes catalyzed tri-deuterium-methylation of H73 in β-actin, whereas no such modification was detected in YPIEHGIVTNWDDMEK peptide (M + H = 1946.9021 Da) or in any other tryptic peptide derived from β-actin protein incubated in the absence of SETD3 (see *Figure 9*).

SETD3 was originally reported as a methyltransferase that methylates histone H3 in zebrafish and mouse (*Kim et al., 2011*; *Eom et al., 2011*). Specifically, the enzyme was shown to methylate a histone H3.3 peptide (H3N4: STGGV**K**), suggesting that it could methylate K36 of mouse histone H3. We therefore compared the SETD3-dependent methylation of actin peptide H (see *Figure 8*) and the histone peptide H3N4. We could see that SETD3 methylated both peptides (*Figure 8—figure supplement 1*), but peptide H was at least 10-fold more efficiently methylated than peptide H3N4, confirming that SETD3 is most probably a histidine *N*-methyltransferase rather than a lysine-specific methyltransferase.

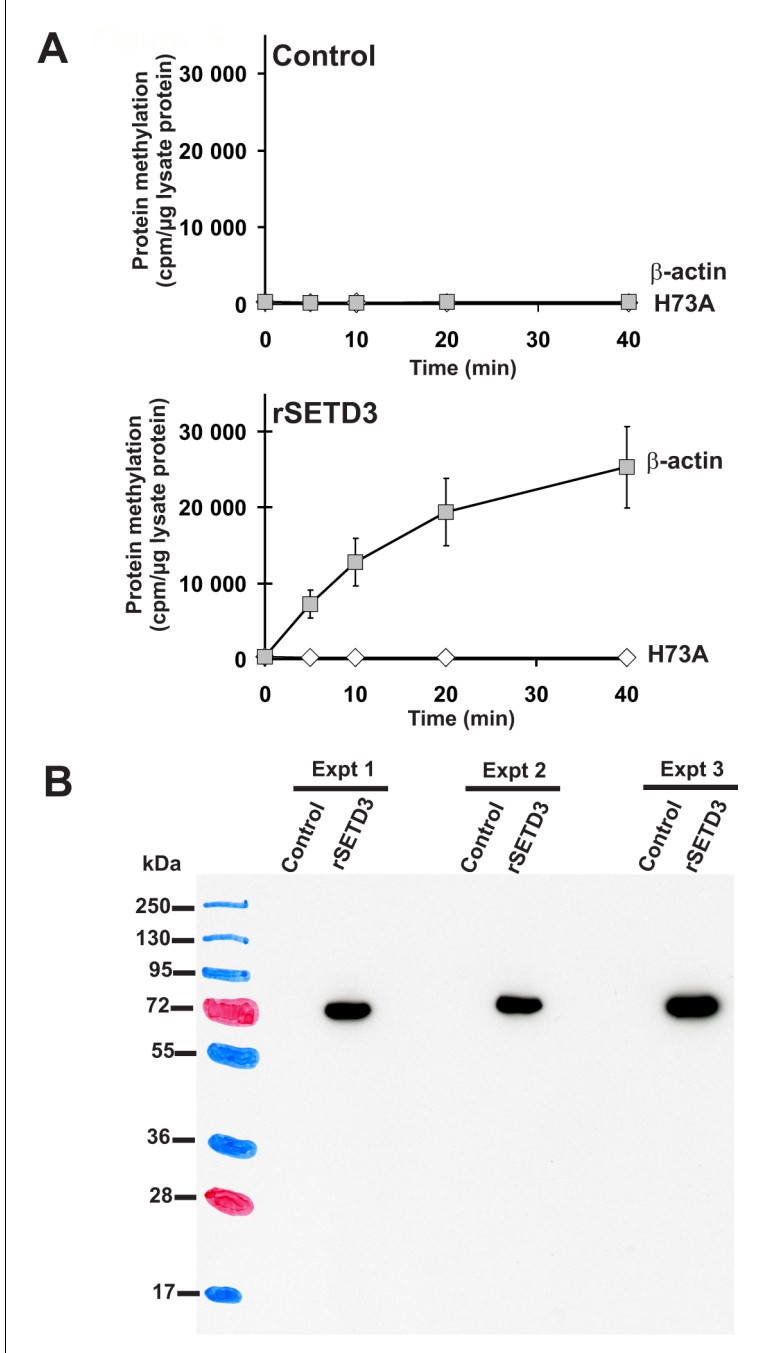

**Figure 5.** Time course of β-actin methylation present in lysates from COS-7 cells that overexpress recombinant rat SETD3. COS-7 cells were transfected for 48 hr with an empty vector (Control) or a vector encoding the rat SETD3 protein (rSETD3). Cells were next harvested and lysed, as described in the 'Materials and methods'. (**A**) The cell-free lysates obtained ($\approx 3$ µg of protein) were incubated at 37°C for the indicated times in the presence of 1 µM (100 pmol, $\approx 370$–$460 \times 10^3$ cpm) [$^1$H+$^3$H]SAM and 2 µM (200 pmol, 8.9 µg) purified recombinant human β-actin (β-actin) or its mutated form (H73A). The reaction was stopped and the proteins present in the assay mixture were precipitated by adding 10% trichloroacetic acid. This allowed for the separation and specific measurement of the radioactivity incorporated in the protein pellet (extent of actin methylation) from the total radioactivity present in the assay mixture. Values are the means ± S.E. (error bars) of three independent transfections. When no error bar is shown, the error is smaller than the width of the line. (**B**) The presence of recombinant SETD3 protein in the cell lysates was verified by Western blot analysis using 20 µg of total protein and an antibody against the His$_6$ tag. The secondary antibody was detected by employing enhanced chemiluminescence and signals acquisition with ECL

*Figure 5 continued on next page*

*Figure 5 continued*
film, whereas the pattern of the prestained protein ladder was copied from the blotting membrane onto the film using a set of felt-tip pens.
DOI: https://doi.org/10.7554/eLife.37921.012
The following source data is available for figure 5:

**Source data 1.** Radiochemical measurements of actin-specific histidine *N*-methyltransferase activity in lysates from COS-7 cells that overexpress recombinant rat SETD3.
DOI: https://doi.org/10.7554/eLife.37921.013

The kinetic properties of recombinant SETD3 proteins were studied in detail using either recombinant human β-actin or the synthetic peptide H as substrates in the presence of *E. coli* *S*-adenosyl-L-homocysteine (SAH) nucleosidase and *Bacillus subtilis* adenine deaminase. The two bacterial enzymes collaborated to break down SAH, a potent inhibitor of methyltransferases, preventing its accumulation in the reaction mixture (*Dorgan et al., 2006*). Rat and human SETD3 have a very high affinity towards both SAM and human β-actin (*Table 3*), with $K_M$ values well below the physiological

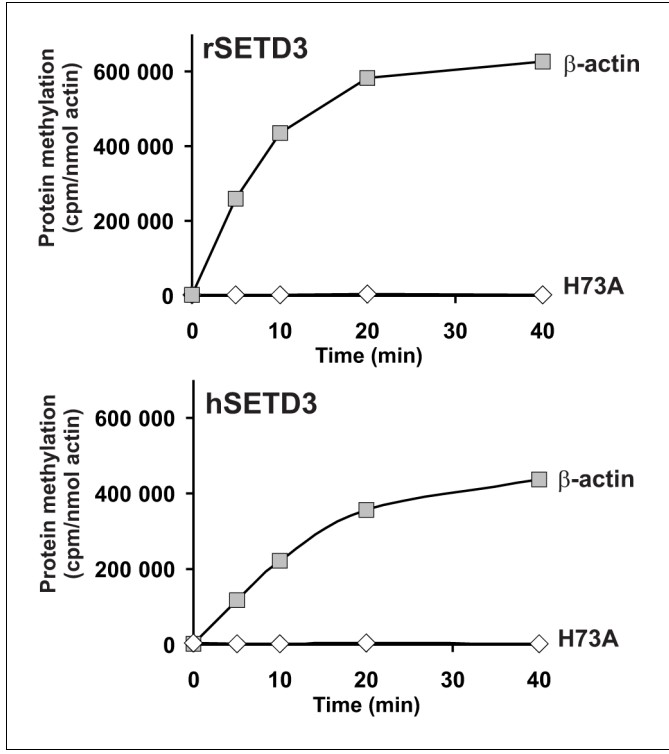

**Figure 6.** In vitro β-actin methylation in the presence of purified recombinant rat or human SETD3 overexpressed in COS-7 cells. Homogeneous recombinant rat SETD3 (rSETD3, 0.4 μg protein) or its human orthologue (hSETD3, 0.3 μg protein) were incubated at 37°C for the indicated times in the presence of 1 μM (100 pmol, ≈ 300–400 × 10³ cpm) [$^1$H+$^3$H]SAM and 2 μM (200 pmol, 8.9 μg) homogenous recombinant human β-actin or its mutated form (H73A). Proteins were precipitated with 10% trichloroacetic acid to determine the incorporation of radioactivity. The figure shows results of representative experiments from two independent experiments performed.
DOI: https://doi.org/10.7554/eLife.37921.014
The following source data and figure supplement are available for figure 6:

**Source data 1.** Radiochemical measurements of β-actin methylation by purified recombinant SETD3 overexpressed in COS-7 cells.
DOI: https://doi.org/10.7554/eLife.37921.016
**Figure supplement 1.** (A) SDS-PAGE and (B) Western-blot analysis of fractions obtained during the purification of the recombinant rat SETD3 protein produced in COS-7 cells.
DOI: https://doi.org/10.7554/eLife.37921.015

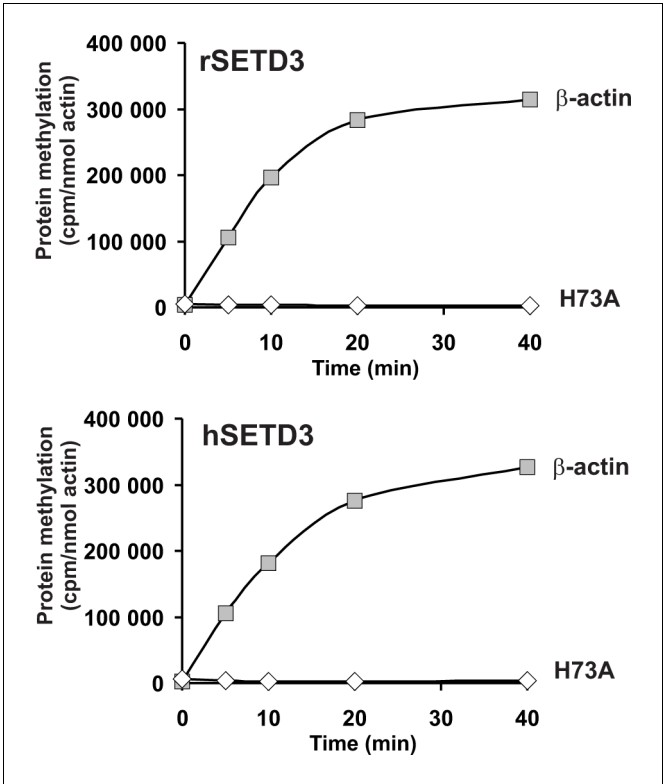

**Figure 7.** In vitro β-actin methylation in the presence of purified recombinant rat or human SETD3 overexpressed in *E. coli*. Mammalian SETD3 proteins were produced in *E. coli* and purified by affinity chromatography on nickel-Sepharose (HisTrap HP), as described in the 'Materials and methods' section. Recombinant rat SETD3 (rSETD3, 0.4 μg protein) or its human orthologue (hSETD3, 0.4 μg protein) were incubated at 37°C for the indicated times in the presence of 1 μM (100 pmol, $\approx 230 \times 10^3$ cpm) [$^1$H+$^3$H]SAM and 2 μM (200 pmol, 8.9 μg) homogenous recombinant human β-actin or its mutated form (H73A). Proteins were precipitated with 10% trichloroacetic acid to determine the incorporation of radioactivity. The figure shows the results of single experiments.

DOI: https://doi.org/10.7554/eLife.37921.017

The following source data and figure supplement are available for figure 7:

**Source data 1.** Radiochemical measurements of β-actin methylation by purified recombinant SETD3 overexpressed in *E. coli*.

DOI: https://doi.org/10.7554/eLife.37921.019

**Figure supplement 1.** (A) SDS-PAGE and (B) Western-blot analysis of fractions obtained during the purification of recombinant human SETD3 produced in *E. coli*.

DOI: https://doi.org/10.7554/eLife.37921.018

concentrations of SAM and β-actin monomers in vertebrate cells ($\approx$ 30 μM and 50–100 μM, respectively (*Clarke and Banfield, 2001*; *Pollard, 2017*). Both enzymes had comparable estimates of $V_{max}$ value at saturating concentrations of SAM, but the $K_M$ for β-actin ($\approx 0.8$ μM) was about 4-fold lower for human SETD3 than for the rat enzyme (3 μM). Consequently, the catalytic efficiency ($K_M/k_{cat}$) of human SETD3 towards β-actin was 3-fold higher than that of the rat enzyme (see *Table 3*). Taking into account the intracellular concentrations of SAM and β-actin monomers, however, the two enzymes probably operate at similar $V_0$ in vivo, i.e. close to their $V_{max}$ ($\approx 10$ nmol.min$^{-1}$.mg$^{-1}$ protein). By contrast, actin peptide H was a very poor substrate for SETD3 as rat and human SETD3 catalyzed methylation of the peptide with extremely low $V_{max}$ values and showed several thousand–fold lower affinity towards the peptide in comparison to the values determined with human β-actin (see *Table 3*).

It is well-established that actin that is produced in bacteria forms non-native mis-folded species that aggregate into inclusion bodies (*Stemp et al., 2005*). This is because actin requires the assistance of the eukaryotic chaperonin CCT (TRiC) to enable it to fold into its native conformation

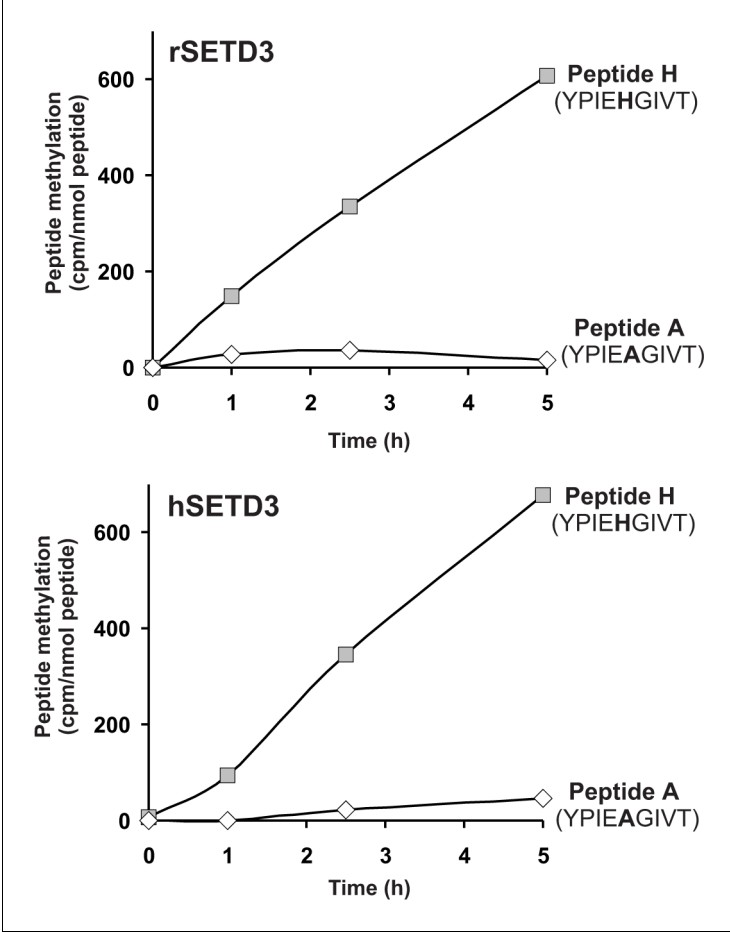

**Figure 8.** SETD3-dependent methylation of synthetic peptides that incorporate the methylation site in β-actin. Homogeneous recombinant rat SETD3 (rSETD3, 5.8 µg protein) or its human orthologue (hSETD3, 4.6 µg protein) was incubated at 37°C for the indicated times in the presence of 1 µM (100 pmol, $\approx$350–400 $\times$ 10$^3$ cpm) [$^1$H+$^3$H] SAM and 2 mM (200 nmol) synthetic peptides corresponding to residues 69–77 of either β-actin (Peptide H: YPIEHGIVT) or its mutated form (Peptide A: YPIEAGIVT). Methylated peptides were separated from [$^1$H+$^3$H]SAM by fractionation on ion exchange Dowex 50W columns. The figure shows the results of single experiments.
DOI: https://doi.org/10.7554/eLife.37921.020

The following source data and figure supplements are available for figure 8:

**Source data 1.** Radiochemical measurements of the SETD3-dependent methylation of synthetic peptides that incorporate the methylation site in β-actin.
DOI: https://doi.org/10.7554/eLife.37921.023

**Figure supplement 1.** SETD3-dependent methylation of synthetic peptide H and peptide H3N4, analogues of the methylation site in β-actin and histone H3, respectively.
DOI: https://doi.org/10.7554/eLife.37921.021

**Figure supplement 1—source data 1.** Radiochemical measurements of SETD3-dependent methylation of synthetic peptides H (b-actin) and H3N4 (histone H3).
DOI: https://doi.org/10.7554/eLife.37921.022

(*Martín-Benito et al., 2002*; *Stemp et al., 2005*). Completely denatured actin is, however, largely capable of re-folding in a spontaneous manner into its native tertiary structure (*Martín-Benito et al., 2002*), and it was this re-folded recombinant β-actin that we used in the experiments described above. We next wanted to know whether STED3 would methylate natively folded actin, like that found in cells, equally well. To this end, recombinant human β-actin was produced in *Saccharomyces cerevisiae*, an eukaryotic species that is devoid of actin-specific histidine methyltransferase activity (*Kalhor et al., 1999*). Surprisingly, neither rat nor human SETD3 methylated ATP-β-

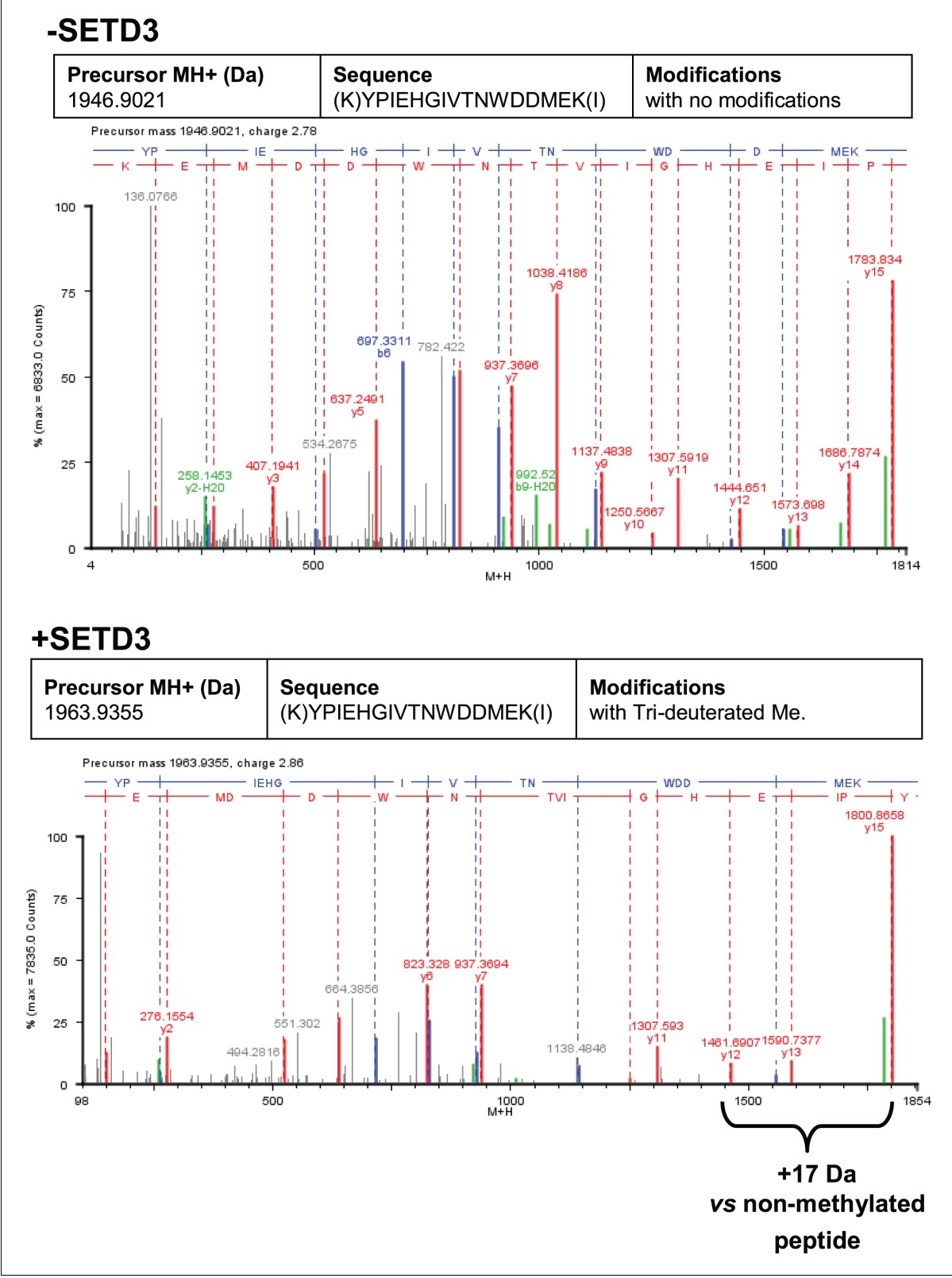

**Figure 9.** Deconvoluted Q-TOF spectra of both non-modified and deuterium($D_3$)-methylated β-actin peptides produced by the human SETD3 protein. Homogenous recombinant human β-actin (6.4 μg) was incubated for 90 min at 37°C in a reaction mixture containing 20–30 μM ([$^2$H]SAM), 4.5–9.0 μg SAH nucleosidase and 1.75–3.5 μg adenine deaminase in either the absence (–SETD3) or presence of 1.1–3.5 μg the homogenous recombinant human SETD3 protein (+SETD3). Following the reaction, β-actin was separated from other proteins by SDS-PAGE and analyzed by tandem mass spectrometry,

*Figure 9 continued on next page*

*Figure 9 continued*

as described in the 'Materials and methods' section. Detection of the trideuterium-methylated peptides was performed by ProteinLynx Global Server 2.4 software (Waters, USA) in a fully automatic mode, and the trideuterium methylation (+1,703,448 Da) of Cys, Asp, Asn, His, Lys, Arg, Glu, Gln residues of tryptic peptides was examined. At least 75% of the actin sequence coverage was achieved in each MS analysis. None of the (trideuterium-) methylated peptides was detected in the control reaction (–SETD3), whereas the H73-containing peptide: YPIEHGIVTNWDDMEK (M + H = 1946.9021 Da, y16 ion) was the only trideuterium-methylated peptide (M + H = 1963.9355 Da, y16 ion) in the presence of SETD3. The mass shift (+17.03448 Da) was detected only in His-containing peptide fragments (y12, y13, y15, y16), indicating that the H73 residue is the site of methylation. The figure shows results of a representative experiment. Two independent methyl(D$_3$)-labeling reactions were performed.
DOI: https://doi.org/10.7554/eLife.37921.024

The following source data is available for figure 9:

**Source data 1.** Mass spectrometry (Q-TOF) identification of both non-modified and deuterium(D3)-methylated β-actin peptides produced in the absence or presence of the human SETD3 protein.
DOI: https://doi.org/10.7554/eLife.37921.025

actin or ADP-β-actin, as determined using the radiochemical assay (*Figure 10*, see *Figures 6* and *7*), suggesting that the endogenous substrate of SETD3 might not be the natively folded nucleotide-bound actin monomers. More importantly, yeast-produced human β-actin became a good substrate for SETD3 when purified in denaturing conditions and re-folded into nucleotide-free quasi-native actin, as did *E. coli*-produced recombinant human β-actin (see *Figure 10*). This effect was not due to the absence of free nucleotides in the reaction mixture because the addition of either 0.5 mM ATP or ADP (a 10-fold excess over the concentrations of nucleotides in the preparation of yeast-produced actin) did not impact the activity of SETD3 towards the refolded actin. To try to solve this puzzle, we prepared nucleotide-free yeast-produced human β-actin monomers in complex with either profilin or cofilin (two well known interacting partner proteins of β-actin [*Lappalainen, 2016*]) and tested whether the actin complexes became good substrates for SETD3. The methylation of profilin-β-actin was negligible and that of cofilin-β-actin complex was low (only ≈3–4% of the total protein in

**Table 3.** Kinetic properties of rat and human SETD3 proteins
Kinetic properties were determined with the use of purified recombinant C-terminal His$_6$-tagged SETD3 protein.

| Substrate | Rat SETD3 | | | | Human SETD3 | | | |
|---|---|---|---|---|---|---|---|---|
| | $V_{max}$ | $K_M$ | $k_{cat}$ | $k_{cat}/K_M$ | $V_{max}$ | $K_M$ | $k_{cat}$ | $k_{cat}/K_M$ |
| | $nmol\ min^{-1}\ mg^{-1}$ | $\mu M$ | $min^{-1}$ | $min^{-1}\ \mu M^{-1}$ | $nmol\ min^{-1}\ mg^{-1}$ | $\mu M$ | $min^{-1}$ | $min^{-1}\ \mu M^{-1}$ |
| β-actin | 11.280 ± 1.018 | 2.996 ± 0.507 | 0.80 | 0.27 | 9.091 ± 0.308 | 0.752 ± 0.070 | 0.65 | 0.86 |
| *S*-adenosyl-L- methionine | 8.053 ± 0.136 | 0.109 ± 0.008 | 0.57 | 5.23 | 8.649 ± 0.119 | 0.116 ± 0.007 | 0.61 | 5.25 |
| Peptide H | 0.064 ± 0.004 | 10590 ± 1373 | 0.005 | $4.7 \times 10^{-7}$ | 0.029 ± 0.003 | 8729 ± 1949 | 0.002 | $2.3 \times 10^{-7}$ |

Determinations for *S*-adenosyl-L-methionine (SAM) were performed with the SETD3 preparations (0.04–0.05 μg protein, 5–7 nM), which were incubated for 8 min at 37°C in the reaction mixture containing 5 μM recombinant β-actin and variable concentrations of [$^1$H+$^3$H] SAM (≈320 × 10$^3$ cpm). The measurements for β-actin were obtained following a 5 min incubation of SETD3 in the presence of 1 μM concentration of [$^1$H+$^3$H] SAM (100 pmol, ≈300 × 10$^3$ cpm). The kinetic parameters of the enzymatic reaction for actin peptide H (YPIEHGIVT) were determined with SETD3 preparations (8.5–9.2 μg protein, 1.2–1.3 μM) incubated for 30 min in the presence of [$^1$H+$^3$H] SAM (100 pmol, ≈290 × 10$^3$ cpm). In all experiments, the reaction mixture contained the homogenous recombinant *S*-adenosyl-L-homocysteine (SAH) nucleosidase (1.6 μg protein, 600 nM, *E. coli*) and adenine deaminase (3.9 μg protein, 600 nM, B. *subtilis*) to prevent SAH accumulation. Values are the means of three or four independent experiments. The values for standard error of the mean (S.E.) are also given.
DOI: https://doi.org/10.7554/eLife.37921.026

The following source data is available for  Table 3:
Source data 1. Determination of the kinetic parameters of SETD3-catalyzed methylation of actin (for actin as the substrate)
DOI: https://doi.org/10.7554/eLife.37921.027

Source data 2. Determination of the kinetic parameters of SETD3-catalyzed methylation of actin (for SAM as the substrate)
DOI: https://doi.org/10.7554/eLife.37921.028

Source data 3. Determination of the kinetic parameters of SETD3-catalyzed methylation of peptide H
DOI: https://doi.org/10.7554/eLife.37921.029

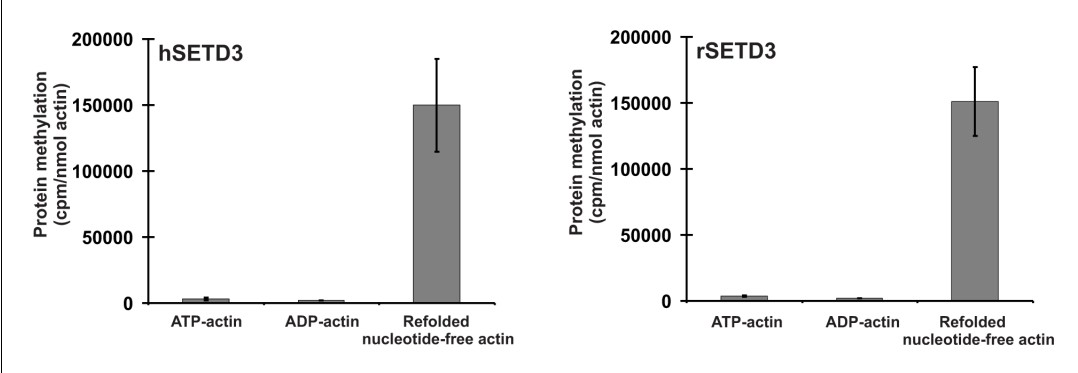

**Figure 10.** SETD3-dependent methylation of human β-actin that is overexpressed in *S. cerevisiae*. Homogeneous recombinant rat SETD3 (rSETD3, 0.6 μg protein) or its human orthologue (hSETD3, 0.45 μg protein) were incubated at 37°C for 40 min in the presence of 1 μM (100 pmol, $\approx 220 \times 10^3$ cpm) [$^1$H+$^3$H]SAM and 2 μM (200 pmol, 8.9 μg) homogenous recombinant human β-actin produced in *S. cerevisiae*, and either purified in non-denaturing conditions as complexes with ATP (ATP-actin) or ADP (ADP-actin) or denatured and refolded to obtain soluble nucleotide-free protein (refolded nucleotide-free actin). Proteins were precipitated with 10% trichloroacetic acid to determine the incorporation of radioactivity. Values are the means ± range (error bars) of two independent measurements performed with two different β-actin preparations.
DOI: https://doi.org/10.7554/eLife.37921.030

The following source data is available for figure 10:

**Source data 1.** Radiochemical measurements of SETD3-dependent methylation of human β-actin overexpressed in *S. cerevisiae*.
DOI: https://doi.org/10.7554/eLife.37921.031

the complex could be methylated). Taken together, our results indicate that globular actin monomers (probably in complex with some actin-binding proteins) or possibly also F-actin can be methylated by SETD3 in vivo.

## SETD3 is the β-actin H73 methyltransferase in HAP1 human cells and *Drosophila melanogaster*

To assess the importance of SETD3 in methylating the β-actin H73 residue in vivo, we used CRISPR/Cas9 technology to inactivate the *Setd3* gene in HAP1 cells. Five distinct *Setd3*-KO clonal HAP1 cell-lines were generated, as described in the 'Materials and methods', and three of these were ultimately selected for further studies (KO-A1, KO-A3 and KO-A5). Sequencing of the DNA showed the presence of mutations, preventing the production of active SETD3 in these cell lines (see 'Materials and methods' for details of the mutations introduced), which was further confirmed by the absence of SETD3 protein in cell extracts from KO-A1, KO-A3 and KO-A5 (*Figure 11*).

To analyze the methylation of H73 in actin from wildtype and SETD3-KO HAP1 cell lines, we partially purified the actin from these lines (*Figure 12A*) and analyzed these proteins by tandem mass spectrometry of the band corresponding to actin from the fraction containing the highest concentration of actin (*Figure 12B*). As shown in *Figure 12C*, methylation of the H73 residue was detected in almost all H73-containing β-actin peptides derived from the wildtype HAP1 cells, whereas close to 90% of the same β-actin peptides derived from two different *Setd3*-deficient cell lines were not methylated at H73. Importantly, the total number of β-actin peptides identified in MS/MS experiments was comparable for wildtype (300–400 Peptide Spectrum Matches) and KO cells ($\approx 500$ Peptide Spectrum Matches), indicating that similar amounts of actin protein were analyzed in all experiments.

We next took advantage of the recently published knock-out model of SETD3 in *D. melanogaster* (*Tiebe et al., 2018*) to investigate the methylation status of actin in vivo. The absence of SETD3 in the fly prevented the methylation of *Drosophila* actin on H74 (which corresponds to the H73 in the mammalian protein). Soluble proteins in homogenates from wildtype and *Setd3*-deficient *Drosophila* larvae were analyzed by tandem mass spectrometry. As shown in *Table 4*, the knock-out of SETD3 prevented the methylation of all three different forms of actin (actin-42A, −57B and −87E) present in extracts from the fly larvae, whereas all H74-containing actin peptides derived from wildtype larvae were methylated.

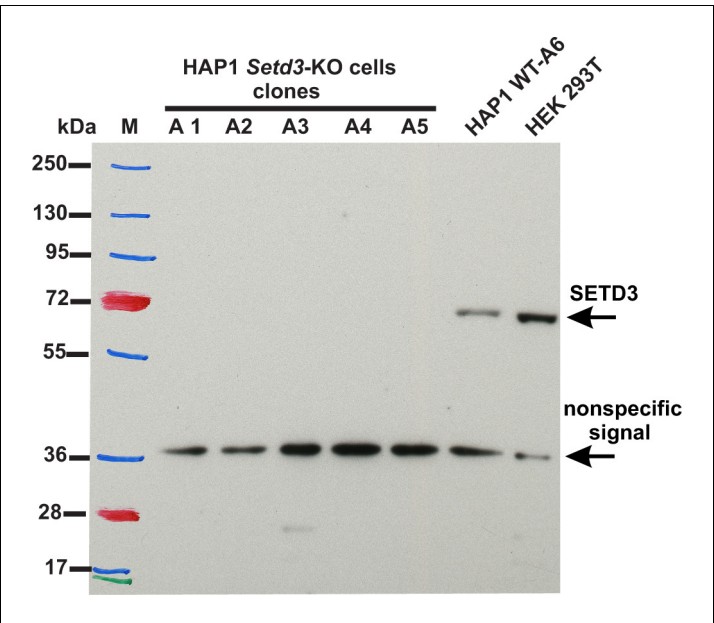

**Figure 11.** Absence of SETD3 expression in *Setd3*-deficient HAP1 cell lines compared to endogenous expression detected in control HAP1 and HEK293T cells. Western blot analysis of SETD3 expression in five *Setd3*-deficient HAP1 clonal cell lines and two different control cell lines: HEK293T and the HAP1 cell line (WT-A6). The HAP1 cell line was also submitted to the CRISPR/Cas9 gene-inactivation procedure, but the *Setd3* gene was not modified; this was done to exclude off-target effects of the procedure on the HAP1 phenotype. The analysis was performed using 50 μg of the cell lysate protein with a rabbit primary antibody against the human SETD3 (ab174662, Abcam) along with a horseradish-peroxidase-conjugated goat anti-rabbit secondary antibody. The secondary antibody was detected by measuring enhanced chemiluminescence. The presence of a nonspecific signal (≈ 36 kDa) is in agreement with the specification of the primary antibody.
DOI: https://doi.org/10.7554/eLife.37921.032

Taken together, we show that actin from wildtype HAP1 cells or *Drosophila* flies was > 90% methylated, whereas in the absence of SETD3, neither human cells nor *Drosophila* can methylate actin H73 or H74, respectively. This confirms the molecular identification of SETD3 as the evolutionarily conserved actin-specific histidine *N*-methyltransferase, and the extent of actin methylation suggests the high functional importance of this post-translational modification.

## Absence of SETD3 reduces F-actin and increases glycolytic rate in SETD3-deficient HAP1 cells

Methylation of the actin H73 residue has been shown to stabilize actin filaments, whereas the absence of such modification resulted in an increased rate of depolymerization of these filaments in vitro (*Nyman et al., 2002*). To determine the effect of *Setd3*-deficiency on the organization of actin filaments, we used confocal microscopy following F-actin staining with TRITC-phalloidin. To obtain a comprehensive picture of actin filaments, the morphology of F-actin was reconstructed in 3D with the use of Zeiss Zen software (Zeiss, Germany). This technique showed a clear decrease in F-actin content in SETD3-KO cells, where actin was largely non-methylated at H73 (*Figure 13*), suggesting an accelerated actin depolymerization and a loss of cytoskeleton integrity. These effects were more pronounced after 48 hr culture (compared to 24 hr), and are in agreement with the results of previous studies using purified recombinant actin (*Nyman et al., 2002*). To exclude artifacts associated with the clonal populations of HAP1 cells, we analyzed two other SETD3-KO clones that showed comparable results (*Figure 13—figure supplement 1*).

Since *Nyman et al. (2002)* showed that the replacement of H73 by an alanine decreased the thermal stability of actin monomers in vitro, this suggested that the methylation of actin at H73 could contribute to its stability, in agreement with our results in HAP1 cells. Therefore, we wanted to compare the in vitro thermostability of actin in cell-free lysates of wildtype (methylated actin) and

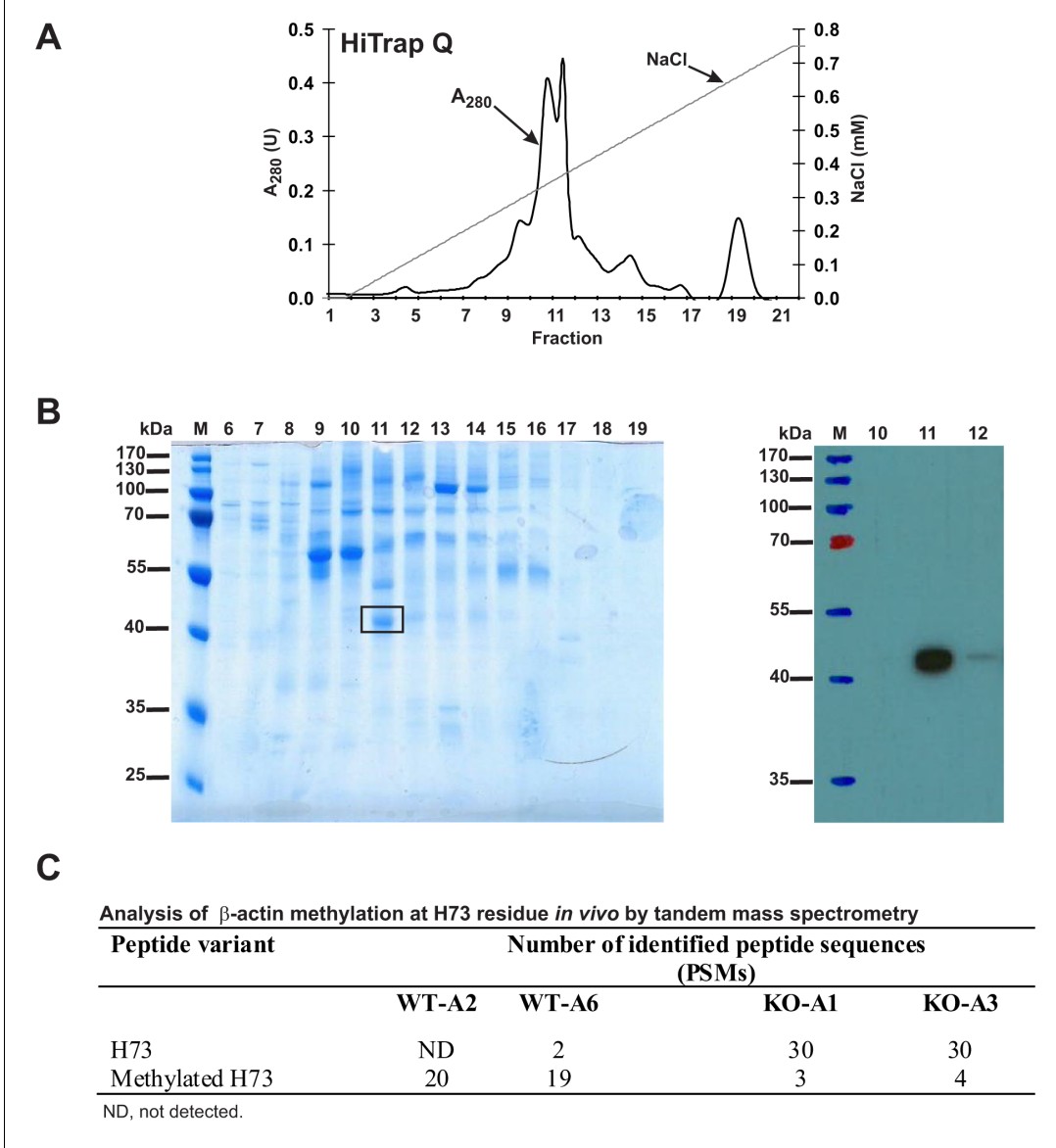

**Figure 12.** Purification of endogenous actin from HAP1 cells and analysis of H73 methylation by tandem mass spectrometry. (**A**) Endogenous actin was purified from *Setd3*-deficient HAP1 cells (clone KO-A1) by chromatography on HiTrap Q HP, as described in the 'Materials and methods' section. (**B**) The indicated fractions that were eluted from the anion-exchange column were analyzed by SDS-PAGE and by western-blotting with the use of an anti-actin primary antibody (A2066, Sigma-Aldrich) to identify protein bands corresponding to the actin protein. M, prestained protein marker. Similar purification and identification procedures were performed with cell lysates from wildtype HAP1 cells (clone WT-A2, WT-A6) and another *Setd3*-deficient HAP1 clone (KO-A3). The indicated band was cut out of the gel, submitted to trypsin digestion, and analyzed by tandem mass spectrometry. (**C**) The table shows the total number of identified peptide sequences (peptide spectrum matches, PSMs) for β-actin that contain either methylated or non-methylated H73 depending on whether the β-actin was isolated from wildtype (WT-A2, WT-A6) or *Setd3*-deficient clonal HAP1 cells (KO-A1, KO-A3). The total number of PSMs recorded for β-actin (Uniprot, P60709) in these MS/MS experiments is as follows: 301 (WT-A2), 387 (WT-A6), 485 (KO-A1) and 529 (KO-A3).

DOI: https://doi.org/10.7554/eLife.37921.033

The following source data is available for figure 12:

**Source data 1.** Mass spectrometry (LTQ XL ion trap) identification of both non-modified and methylated β-actin peptides from HAP1 human cells.
DOI: https://doi.org/10.7554/eLife.37921.034

**Table 4.** Analysis of actin methylation at the H74 residue in the tissues of *Drosophila* larvae by tandem mass spectrometry.

The total number of identified peptide sequences with H74 (peptide spectrum matches, PSMs) for three different forms of actin present in either wildtype *Drosophila* larvae (WT) or *Setd3* KO animals (SETD3 KO) are indicated. The total numbers of PSMs recorded for actin proteins in these MS/MS experiments are as follows: 90 (Actin-42A, WT), 118 (Actin-57B, WT), 111 (Actin-87E, WT), 86 (Actin-42A, SETD3 KO), 111 (Actin-57B, SETD3 KO), 78 (Actin-87E, SETD3 KO).

| Actin | Peptide variant | Number of identified peptide sequences (PSMs) | |
|---|---|---|---|
| | | WT | SETD3 KO |
| Actin-42A (P02572) | H74 | 0 | 14 |
| | Methylated H74 | 12 | 0 |
| Actin-57B (P53501) | H74 | 0 | 19 |
| | Methylated H74 | 9 | 0 |
| Actin-87E (P10981) | H74 | 0 | 19 |
| | Methylated H74 | 9 | 0 |

DOI: https://doi.org/10.7554/eLife.37921.035

The following source data is available for Table 4:

Source data 1. Mass spectrometry (Orbitrap Fusion Lumos tribrid) identification of both non-modified and methylated actin peptides from the tissues of *Drosophila melanogaster* larvae.
DOI: https://doi.org/10.7554/eLife.37921.036

SETD3-KO (non-methylated actin) HAP1 cells. When cell extracts were heated at increasing temperatures, it became clear that the absence of methylation of H73 destabilized actin in the temperature range between 30°C to 46°C (**Figure 14**). Taken together, these results suggest that the loss of methylation in H73 leads to a structural instability of actin monomers that might, in turn, result in the accelerated depolymerization of actin filaments in SETD3-KO cells.

A routine daily inspection of HAP1 cell cultures revealed that at about 80–90% culture confluency, *Setd3*-deficient HAP1 cells tended to acidify Iscove's Modified Dulbecco's Medium (IMDM) faster than the control cells. This phenotype was not attributable to the differences in cell culture density between control and SETD3-KO cell lines, suggesting an increased formation of lactate from glucose by *Setd3*-deficient cells. To test this possibility, both control and *Setd3*-deficient HAP1 cells (90% culture confluency) were further incubated in fresh IMDM medium for 20 hr, after which the medium was collected, and glucose consumption and lactate production were measured in a neutralized $HClO_4$ extract. As shown in **Figure 15**, both the formation of lactate and the consumption of glucose in SETD3-KO cell lines were increased by about 15–20% in comparison with those determined for the control cells, indicating a shift of their metabolism towards a more glycolytic phenotype. Taken together, these results suggest that the lack of methylation at H73 in HAP1 cells destabilizes F-actin, which possibly explains the increase in glucose consumption that occurs in these cells.

## Discussion

### Molecular identity of rat actin-specific histidine *N*-methyltransferase

The presence of two distinct histidine *N*-methyltransferases in rabbit leg muscle has been reported by *Raghavan et al. (1992)*. The first enzyme, carnosine *N*-methyltransferase catalyzes the *N*π-methylation of the histidine imidazole ring of carnosine (β-alanyl-L-histidine), an abundant dipeptide in the skeletal muscle of vertebrates. The second enzyme was shown to methylate the *N*τ position of the H73 residue of mammalian β-actin protein. The previous studies performed in our laboratory resulted in the identification of genes coding for carnosine *N*-methyltransferase in vertebrates (*Drozak et al., 2013*; *Drozak et al., 2015*), whereas the current investigation presents data that reveal the molecular identification of rat actin-specific histidine *N*-methyltransferase as the SET domain-containing protein 3 (SETD3), disclosing the molecular identity of the mammalian actin-

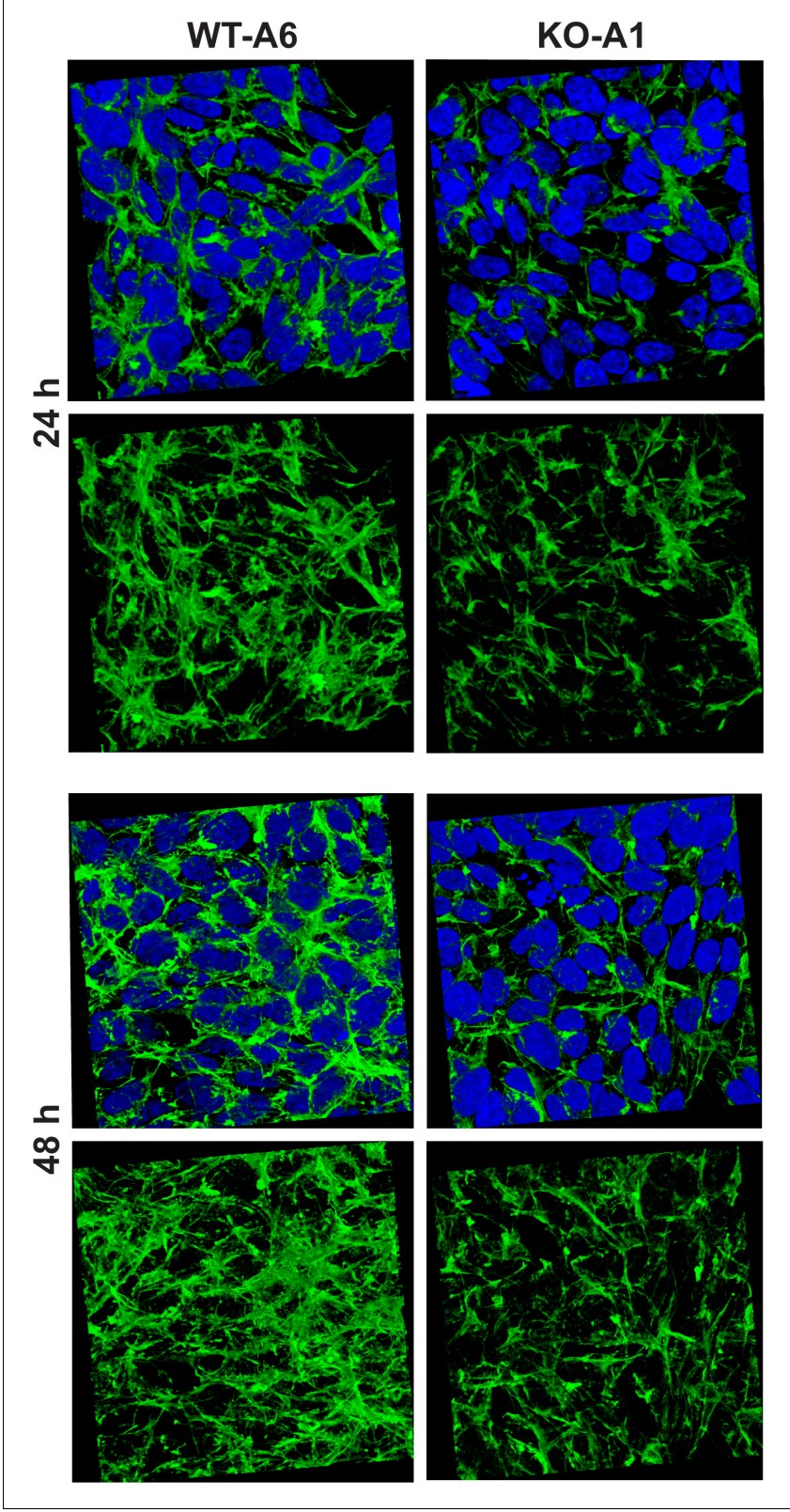

**Figure 13.** Organization of the actin cytoskeleton in *Setd3*-deficient HAP1 cells visualized by confocal microscopy. Control HAP1 cells (WT-A6) and Setd3-deficient clonal HAP1 cells (KO-A1) were cultured for either 24 or 48 hr. The localization of filamentous actin was determined by staining with TRITC-phalloidin (green), while nuclei were visualized by staining with Hoechst 33342 (blue). Three-dimensional confocal microscope images of actin

*Figure 13 continued on next page*

*Figure 13 continued*
cytoskeleton are shown with or without stained nuclei (blue). For 3D analysis, confocal Z-stacks comprising 10–13 optical slices were reconstructed into each 3D image with the aid of Zeiss Zen software. Representative images are shown here, but similar results were obtained with two other HAP1 *Setd3*-deficient clonal cell lines (KO-A3 and KO-A5, see *Figure 13—figure supplement 1*).
DOI: https://doi.org/10.7554/eLife.37921.037
The following figure supplement is available for figure 13:

**Figure supplement 1.** Organization of the actin cytoskeleton in *Setd3*-deficient HAP1 cells visualized by confocal microscopy.
DOI: https://doi.org/10.7554/eLife.37921.038

methylating enzyme. This conclusion is supported by the following findings: (i) two different and extensive rounds of purification of the actin-methylating activity from rat skeletal muscle resulted in the identification of the protein SETD3 as the only logical candidate matching the enzymatic activity; (ii) recombinant rat SETD3 catalyzes the specific methylation of the H73 residue present in both the recombinant β-actin protein and the synthetic peptide corresponding to the nucleotide sensor loop of β-actin; (iii) in contrast to actin isolated from control HAP1 cells, the protein present in *Setd3*-deficient cells is essentially devoid of methylated H73 residues, as verified by MS/MS analysis; and (iv) the inactivation of SETD3 in *Drosophila Setd3* knockout larvae results in the complete absence of H74 methylation in all three different actins found in *Drosophila*.

The methylation of H73 is a highly conserved modification among actins from many different eukaryotes, although it has been found to be absent from the actin of the amoeboflagellate *N. gruberi* and the yeast *S. cerevisiae* (*Sussman et al., 1984*; *Nyman et al., 2002*). Orthologues of the protein SETD3 are present in animals, plants and certain fungi, but not in the proteomes of the abovementioned proteozoan and yeast species, further supporting the identification of SETD3 as the actin-specific histidine *N*-methyltransferase.

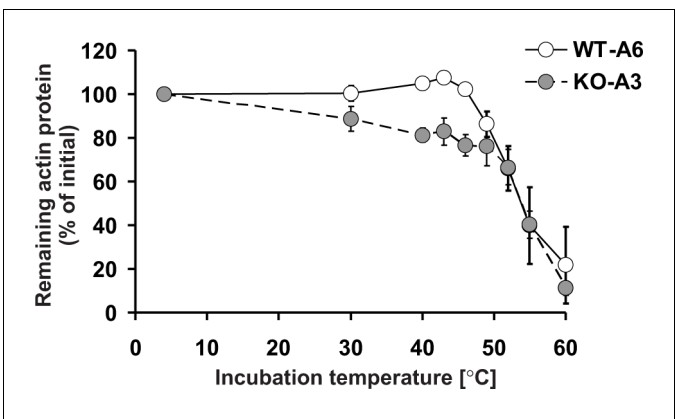

**Figure 14.** Thermal stability of actin in *Setd3*-deficient HAP1 cells. Cell free lysates of control HAP1 cells (WT-A6) and Setd3-deficient clonal HAP1 cells (KO-A3) were incubated for 5 min at 4, 30, 40, 43, 46, 49, 52, 55 and 60°C. Precipitated proteins were then pelleted, and the resulting supernatants (24 or 48 μg protein) were analyzed by SDS-PAGE and by western-blotting, with the use of anti-actin primary antibody (A2066, Sigma-Aldrich) to visualize protein bands corresponding to the actin protein. The levels of actin were quantified by densitometry using Quantity One software (BioRad) and normalized to the intensity of the actin band of the sample incubated at 4°C. Values are the means ± S.E. (error bars) of three independent experiments. When no error bar is shown, the error is smaller than the width of the line.
DOI: https://doi.org/10.7554/eLife.37921.039
The following source data is available for figure 14:

**Source data 1.** Quantification of western blots with Quantity One software.
DOI: https://doi.org/10.7554/eLife.37921.040

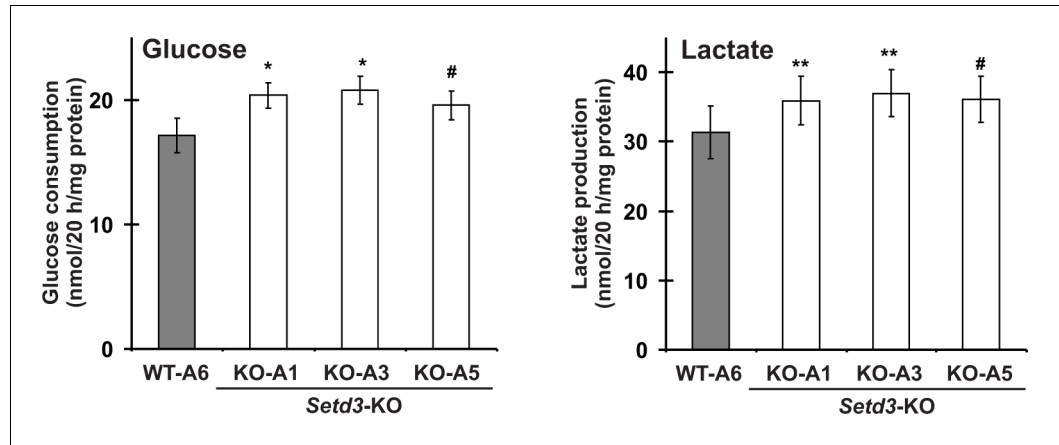

**Figure 15.** Impact of SETD3 inactivation on glucose consumption and lactate production in HAP1 cells. Experiments were performed on three HAP1 cell lines that were deficient in *Setd3* (KO-A1, KO-A3 and KO-A5) and a control HAP1 cell line (WT-A6) that has been submitted to the CRISPR/Cas9 gene inactivation procedure, but yielding no modification of the *Setd3* gene. HAP1 cells (≈90% culture confluency) were supplemented with fresh medium and incubated for 20 hr as described in the 'Materials and methods'. The glucose and lactate concentrations in samples of deproteinized culture medium were determined spectrophotometrically as described in the 'Materials and methods'. Values are the means ± S.E. (error bars) of three independent experiments performed with cells from three different culture passages (n = 3). The data were assumed to be distributed normally, and the homogeneity of variances was verified with F tests. Statistical significance was analyzed using a one-tailed paired Student's t-test (GraphPad Prism 4.0). [#]$p<0.09$, [*]$p<0.05$, [**]$p<0.01$.
DOI: https://doi.org/10.7554/eLife.37921.041

The following source data is available for figure 15:

**Source data 1.** Quantification of glucose consumption in HAP1 human cells.
DOI: https://doi.org/10.7554/eLife.37921.042
**Source data 2.** Quantification of lactate production in HAP1 human cells.
DOI: https://doi.org/10.7554/eLife.37921.043

SETD3 was originally reported as a methyltransferase that catalyzes the modification of histone H3 at K4 and K36 residues in zebrafish and mouse (*Kim et al., 2011*; *Eom et al., 2011*). The independent studies of *Chen et al., 2013* confirmed the histone methyltransferase activity of SETD3 and suggested that the enzyme may have other non-histone substrates in the cytoplasm owing to the presence of the Rubisco LSMT substrate-binding domain in its structure. This view was further strengthened by the finding that SETD3 is predominantly localized in the cytosol of human cells, while only a small portion is found in the nucleus (*Cheng et al., 2017*). In the present study, we have not examined the activity of recombinant SETD3 towards histones because isolated histones are extremely poor substrates (*Chen et al., 2013*). When we investigated the methyltransferase activity of SETD3 with either the actin peptide H or the histone peptide H3N4, however, the activity with the histone peptide was at least a 4000-fold lower than that of recombinant β-actin. Consequently, we identified actin as the novel physiological substrate of SETD3, which is present in the cytoplasm and is likely to be the most abundant actin methyltransferase. It is possible however, that SETD3 may also be a dual methyltransferase able to methylate both the Lys and His residues of other proteins, but this still requires in vivo confirmation.

## Quasi-native β-actin from bacterial inclusion bodies is a good substrate for the actin-specific histidine *N*-methyltransferase

The molecular identification of the actin-specific histidine *N*-methyltransferase through its extensive purification from rat muscles was only possible because we were able to prepare large amounts of pure β-actin monomers to be used as the substrate for measuring the enzymatic activity in radio-chemical assays. Although this β-actin was produced and subsequently purified from the inclusion bodies of *E. coli* in the nucleotide-free form, it remained a good substrate for the methyltransferase. Given that, on the one hand, it is not possible to produce natively folded actin in bacteria

(*Stemp et al., 2005*; *Martín-Benito et al., 2002*) and that, on the other hand, the ATP/ADP-free actin denatures rapidly and irreversibly (*Graceffa and Dominguez, 2003*), it is likely that the actin preparation used in our study may comprise actin molecules with stable non-native spatial structures. Indeed, the actin methylation reaction was still substoichiometric, and we have observed the methylation of up to 35% of the total actin present in the reaction mixture. This observation was readily reproducible in the presence of bacterial SAH nucleosidase and adenine deaminase, suggesting that it was not due to a potential accumulation of SAH in the reaction mixture (unpublished data, Drozak et al.). This indicates that only a small portion of the actin monomers produced in bacteria and purified under denaturing conditions adopted a conformation that was recognized by the SETD3 enzyme. This observation is in line with a previous report, showing only a minor methylation of actin inclusion bodies (up to 1% protein) by rabbit actin methyltransferase (*Raghavan et al., 1992*). Furthermore, SETD3 was at least 200-fold less active towards the synthetic peptide H, corresponding to the amino acid sequence surrounding the H73, than was recombinant β-actin (see *Table 3*). This indicated that structural features of the actin molecule other than the primary sequence of the sensor loop are probably important for its recognition by the actin methyltransferase.

Both tested SETD3 orthologs are rather 'lazy' enzymes, with a $k_{cat}$ of 0.6–0.8 min$^{-1}$, resembling the values estimated for protein methyltransferases rather than those measured for enzymes catalyzing the methylation of low-molecular-weight substrates (*Clarke and Banfield, 2001*; *Osborne et al., 2007*). Considering that the intracellular concentrations of SAM and β-actin monomers in vertebrate cells are about 30 μM and 50–100 μM, respectively (*Clarke and Banfield, 2001*; *Pollard, 2017*), both rat and human SETD3 have a high affinity for both SAM ($K_M \approx$ 0.3 μM) and β-actin ($K_M \approx$ 0.8–3 μM). These observations suggest that the enzymes may be fully saturated by substrates and operate at rates that are close to their $V_{max}$ in vivo. It is important to keep in mind, however, that the recombinant β-actin used in the present kinetic studies is certainly not a physiological form of β-actin. Therefore, the affinity of SETD3 to physiological β-actin remains to be verified.

It is very intriguing to note that in our hands both the recombinant ATP- and ADP-β-actins produced in yeast (*S. cerevisiae*), which are expected to have a native conformation, were not methylated by the SETD3 enzyme. By contrast, the yeast-produced actin that was purified in denaturing conditions to yield nucleotide-free quasi native protein became a good substrate for the methyltransferase. This finding may suggest that the presence of ATP or ADP could create a structural hindrance that prevents SETD3 from reaching the H73 residue if the methyltransferase interacts with actin through the nucleotide-binding cleft (see *Figure 4*).

Taken together, these observations have led us to hypothesize that nucleotide-free actin monomers in a complex with one or more actin-binding proteins of yet unknown identity could be the physiological substrate for the SETD3 methyltransferase. If so, this resembles the activity of the yeast Yil110w protein, the only other known protein histidine *N*-methyltransferase, which methylates His243 of the ribosomal Rpl3 protein only when Rpl3 is associated with the ribosome (*Webb et al., 2010*; *Al-Hadid et al., 2014*).

This hypothesis fits with the current knowledge on the enzymes belonging to the family of SET-domain-containing proteins. Indeed, many of these methyltransferases are found in complexes with numerous other proteins that are essential for their catalytic activity and substrate specificity. For example, to exert its catalytic activity, the yeast histone H3 methyltransferase SET1 requires the presence of seven different proteins that form a complex named COMPASS (Complex of Proteins Associated with Set1), and similar COMPASS-like complexes are formed by other mammalian SET-domain-containing methyltransferases (for a review, see *Herz et al., 2013*).

## Biological importance of the SETD3 methyltransferase

Several reports describing the role of SETD3 in the regulation of the cell cycle and apoptosis (*Kim et al., 2011*), myocyte differentiation (*Eom et al., 2011*), cell response to hypoxic conditions (*Cohn et al., 2016*) and tumorigenesis (*Pires-Luís et al., 2015*; *Cheng et al., 2017*) have been published to date. Some of these studies identified the methylation of histones as a primary mechanism of SETD3 action (*Kim et al., 2011*; *Eom et al., 2011*), whereas others suggested that the enzyme may instead modify non-histone substrates that are present in the cytosol (*Chen et al., 2013*; *Cohn et al., 2016*). The existence of two distinct domains in SETD3, i.e. SET and Rubisco LSMT

substrate-binding domains, has been suggested to be the structural basis for such dual substrate specificity (*Chen et al., 2013*).

Our results identify β-actin as a novel substrate of SETD3 and show that disruption of its activity leads to the depletion of F-actin and a loss of cytoskeleton integrity. This observation is probably explained by an increased rate of depolymerization of the non-methylated actin filaments. As a result, there is an accelerated conversion of a more structurally stable ATP-F-actin into a less stable ADP-F-actin, as also shown for a hypomethylated form of recombinant actin in vitro (*Nyman et al., 2002*). Thus, the biochemical role of the methylated H73 residue is likely to contribute to stabilizing the structure of monomers that build the ATP-F-actin and to delay the intermolecular conformational rearrangements of F-actin that complement the hydrolysis of ATP. This conclusion finds at least partial confirmation in the thermal stability assay, which shows that the methylated actin derived from the wildtype HAP1 cells is clearly more thermo-stable than the non-methylated one (SETD3-KO cells) (see *Figure 14*). Taken together, our results show that the phenotype that we have identified in *Setd3*-deficient human HAP-1 cells is caused by the absence of the catalytic activity of SETD3, rather than by a non-catalytic or actin-stabilizing interaction between SETD3 and actin filaments.

The inherent instability of F-actin in *Setd3*-deficient HAP1 cells is likely to be at the origin of the increased lactate production by these cells. Both the polymerization and the depolymerization of actin filaments occur continuously even in resting cells and constitute a major energy drain, causing up to 50% of total ATP consumption in vertebrate cells (*Kudryashov and Reisler, 2013*). Thus, the destabilization that leads to the accelerated degradation of hypomethylated F-actin probably drives an increasing demand for ATP that could be fulfilled by shifting the cell metabolism towards glycolysis. The phenotype of *Setd3*-deficient cells is therefore somewhat similar to that of malignant cells, suggesting that SETD3 activity may play a role in suppressing tumor development. Indeed, the hypomethylation of actin was observed in rat cells following their transformation with the Src oncogen (*Chiou et al., 2012*). Furthermore, the enzymatically inactive SET-domain-loss form of SETD3 was shown to function as a dominant-negative mutant, promoting oncogenesis in human cells (*Chen et al., 2013*), most probably by blocking the methylation of an unknown SETD3 substrate. It is tempting to speculate that actin could be this substrate.

## Conclusions

In the current investigation, we have identified the rat actin-specific histidine *N*-methyltransferase as SETD3, an enzyme catalyzing the extremely well-conserved methylation of H73 in β-actin. The SETD3 enzyme is therefore the first protein-histidine methyltransferase to be identified in vertebrates and the first SET-domain-containing enzyme possibly displaying a dual methyltransferase specificity towards both Lys and His residues in proteins. SETD3 is responsible for the methylation of H73 in human and *Drosophila* actin, and a loss of SETD3 activity in human HAP1 cells induces phenotypic changes resembling those present in cancer cells, suggesting that the hypomethylation of actin might be involved in tumorigenesis. Finally, this work also shows that non-native protein substrates might be useful tools in the search for novel protein methyltransferases.

## Note added in proof

While this paper was in production, similar and complementary data describing the identification of SETD3 as the actin-specific histidine *N*-methyltransferase was accepted for publication (*Wilkinson et al., 2018*). Since the work was independently carried out by both research groups and equally contributes to reveal the biochemical importance of SETD3 enzyme as the actin-specific H73 N-methyltransferase, we ask the authors referring to the function SETD3 in their future publications to mention both this paper, as well as *Wilkinson et al. (2018)*.

# Materials and methods

**Key resources table**

| Reagent type (species) or resource | Designation | Source or reference | Identifiers | Additional information |
|---|---|---|---|---|

*Continued on next page*

*Continued*

| Reagent type (species) or resource | Designation | Source or reference | Identifiers | Additional information |
|---|---|---|---|---|
| Antibody | Rabbit anti-SETD3 antibody | Abcam | ab174662; RRID: AB_2750852 | (1:10,000) |
| Antibody | Rabbit anti-Actin antibody | Sigma-Aldrich | A2066; RRID: AB_476693 | (1:1000) |
| Antibody | Horseradish peroxidase-conjugated goat anti-rabbit IgG antibody | Agrisera | AS09602; RRID: AB_1966902 | (1:20,000; 1:25,000) |
| Antibody | Mouse anti-His6 tag antibody | GE Healthcare; PMID: 23705015 | 27-4710-01; RRID: AB_771435 | (1:2000) |
| Antibody | Horseradish peroxidase-conjugated goat anti-mouse antibody | Sigma-Aldrich; PMID: 23705015 | A2554; RRID: AB_258008 | (1:10,000) |
| Cell line (*Homo sapiens*) | HAP1 | Horizon Discovery | C859; RRID:CVCL_Y019 | The HAP1 cell line was bought directly from Horizon Discovery (Waterbeach, UK) and has been control quality checked by the vendor |
| Cell line (*Cercopithecus aethiops*) | COS-7 | Cell Lines Service | 605470; RRID: CVCL_0224 | The COS-7 cell line was bought directly from CLS (Eppelheim, Germany) and has been control quality checked by the vendor |
| Gene (*Escherichia coli*) | SAH nucleosidase | NA | NCBI: NC_000913.3 | |
| Gene (*Bacillus subtilis*) | Adenine deaminase | NA | NCBI: NC_000964.3 | |
| Gene (*H. sapiens*) | β-actin | NA | GenBank: NM_001101.4 | |
| Gene (*H. sapiens*) | SETD3 | NA | NCBI: NM_032233.2 | |

*Continued on next page*

Continued

| Reagent type (species) or resource | Designation | Source or reference | Identifiers | Additional information |
|---|---|---|---|---|
| Gene (*Rattus norvegicus*) | SETD3 | NA | NCBI: XM_002726774.2 | |
| Gene (*H. sapiens*) | Cofilin-1 | NA | NCBI: NM_005507.2 | |
| Gene (*H. sapiens*) | Profilin-1 | NA | NCBI: NM_005022.3 | |
| Recombinant DNA reagent | pCOLD I (plasmid) | Takara Bio | 3361 | |
| Recombinant DNA reagent | pESC-URA (plasmid) | Agilent Technologies | 217454 | |
| Recombinant DNA reagent | pEF6/Myc -His A (plasmid) | Invitrogen | V96220 | |
| Recombinant DNA reagent | pSpCas0n (BB)—2A-Puro (plasmid) | Other | | Kind gift of F. Zhang, Massachusetts Institute of Technology |
| Genetic reagent (*C. aethiops*) | pEF6/SETD3 | This paper | | Transient overexpression of the recombinant human and rat SETD3 proteins in the COS-7 cell line |
| Genetic reagent (*E. coli*) | pCOLD I/SETD3 | This paper | | Overexpression of the recombinant human and rat SETD3 proteins in the *E. coli* BL21 (DE3) |
| Genetic reagent (*E. coli*) | pCOLD I/β-actin | PMID: 20851184 | | Kind gift of M. Tamura, Ehime University |
| Genetic reagent (*E. coli*) | pCOLD I/SAH nucleosidase | This paper | | Overexpression of the recombinant *E. coli* SAH nucleosidase in the *E.coli* BL21 (DE3) |

*Continued*

| Reagent type (species) or resource | Designation | Source or reference | Identifiers | Additional information |
|---|---|---|---|---|
| Genetic reagent (*E. coli*) | pCOLD I/adenine deaminase | This paper | | Overexpression of the recombinant *B.subtilis* adenine deaminase in the *E. coli* BL21 (DE3) |
| Genetic reagent (*E. coli*) | pCOLD I/Cofilin-1 | This paper | | Overexpression of the recombinant human cofilin-1 in the *E. coli* BL21 (DE3) |
| Genetic reagent (*E. coli*) | pCOLD I/Profilin-1 | This paper | | Overexpression of the recombinant human profilin-1 in the *E. coli* BL21 (DE3) |
| Genetic reagent (*Saccharomyces cerevisiae*) | pESC-URA/β-actin | This paper | | Overexpression of the recombinant human β-actin in the *S. cerevisiae* BY4742 strain |
| Strain, strain background (*B. subtilis*) | *Bacillus subtilis* | Sigma-Aldrich | ATCC:6633 | |
| Strain, strain background (*E. coli*) | *E. coli* BL21 (DE3) | Agilent Technologies | 200131 | |
| Strain, strain background (*S. cerevisiae*) | BY4742 | EUROSCARF | Y10000 | |
| Strain, strain background (*S. cerevisiae*) | BY4742/β-actin | This paper | | BY4742 strain overexpressing recombinant human β-actin |
| Peptide, recombinant protein | Peptide H | Caslo Laboratory | | |
| Peptide, recombinant protein | Peptide A | Caslo Laboratory | | |
| Peptide, recombinant protein | Peptide H3N4 | Caslo Laboratory | | |
| Chemical compound, drug | *S*-[methyl-$^3$H]adenosyl-L-methionine; [$^3$H]SAM | PerkinElmer | NET155V250UC; NET155V001MC | |

*Continued on next page*

*Continued*

| Reagent type (species) or resource | Designation | Source or reference | Identifiers | Additional information |
|---|---|---|---|---|
| Chemical compound, drug | S-[methyl-$^2$H] adenosyl-L-methionine; [$^2$H]SAM | C/D/N Isotopes Inc. | D-4093 | |
| Software, algorithm | ProteinLynx Global Server 2.4; PLGS 2.4 | Waters | RRID: SCR_016664 | |
| Software, algorithm | Quantity One | BioRad | RRID: SCR_016622 | |
| Software, algorithm | GraphPad Prism 4.0 | GraphPad Software | RRID: SCR_002798 | |
| Software, algorithm | Proteome Discoverer | Thermo Fisher Scientific | RRID: SCR_014477 | |
| Software, algorithm | Zeiss Zen | Zeiss | RRID: SCR_013672 | |

## Materials

DEAE-Sepharose, Q-Sepharose, Phenyl-Sepharose, Superdex 200, HiScreen Blue FF, HisTrap HP (Ni$^{2+}$ form) and PD-10 columns were obtained from GE Healthcare Life-Sciences (Little Chalfont, UK). Reactive Red 120-Agarose and Dowex 50W-X4 (200 mesh) resins came from Sigma-Aldrich, and Vivaspin-500 and 20 centrifugal concentrators were from Sartorius Stedim (Goettingen, Germany). All other enzymes and DNA-modifying enzymes as well as the TurboFect transfection reagent were obtained from Thermo-Fermentas (Waltham, USA), A and A Biotechnology (Gdynia, Poland) or Bio-Shop (Burlington, Canada).

## Assay of the actin-specific histidine *N*-methyltransferases activity

### Protein substrates

Enzyme activity was determined by measuring the incorporation of the [$^3$H]methyl group from S-[methyl-$^3$H]adenosyl-L-methionine ([$^3$H]SAM) into homogenous recombinant human (mammalian) β-actin or its mutated form in which histidine 73 was replaced by an alanine residue (H73A). The standard incubation mixture (0.11 ml) contained 25 mM Tris-HCl, pH 7.2, 10 mM KCl, 1 mM DTT, 2 μM protein substrate and 1 μM [$^1$H+$^3$H] SAM ($\approx 400 \times 10^3$ cpm). When appropriate, the incubation mixture was supplemented with recombinant S-adenosyl-L-homocysteine (SAH) nucleosidase and adenine deaminase, as indicated in the legends to the figures and tables. Both auxiliary enzymes were added in $\approx$ 100-fold molar excess in comparison to SETD3. The reaction was started by the addition of the enzyme preparation and carried out at 37°C for 15 min unless otherwise described. Protein methylation was linear for at least 15 min under all of the conditions studied. By analogy to assays of nonribosomal peptide synthetase activity (*Richardt et al., 2003*; *Drozak et al., 2014*), the incubation was stopped by the addition of 0.1 ml of the reaction mixture to 0.025 ml of bovine serum albumin (BSA) (1 mg) and 0.8 ml of ice-cold 10% (w/v) trichloroacetic acid (TCA). After 10 min on ice, the precipitate was pelleted and washed twice with ice-cold 10% TCA. The pellet was finally dissolved in pure formic acid.

### Peptide substrates

The enzyme-dependent methylation of synthetic peptides — Peptide H (YPIEHGIVT) and Peptide A (YPIEAGIVT), analogs of methylation sites in mammalian β-actin, or peptide H3N4 (STGGVK), corresponding to a putative methylation site in mouse histone H3.3 — was determined by measuring the incorporation of the [$^3$H]methyl group from [$^3$H]SAM into a corresponding peptide. The composition of a standard incubation mixture was similar to that described above, excepting that protein substrates were replaced by peptides at 2 mM concentration. Blanks containing no peptides were included in all assays. The reaction was started by the addition of enzyme preparation and carried out at 37°C for the time periods indicated in the figure captions. The incubation was stopped by the addition of 0.1 ml of the reaction medium to 0.2 ml of ice-cold 10% (w/v) HClO$_4$. The samples were

diluted with 0.12 ml of $H_2O$ and centrifuged at 13,000 $\times$ g for 10 min. After neutralization of the supernatant with 3 M $K_2CO_3$/3 M KOH, the salts were removed by centrifugation (13,000 $\times$ g for 10 min) and the clear supernatant was diluted five times with 20 mM Hepes, pH 7.5. 2 ml of the diluted supernatant was applied to Dowex 50W-X4 columns (1 ml, $Na^+$ form) and equilibrated with 20 mM Hepes, pH 7.5. The columns were washed with 3 $\times$ 2 ml of 20 mM Hepes pH 7.5, and the resulting fractions contained (non)methylated peptides H and A. Methylated peptide H3N4 was eluted with 3 $\times$ 2 ml of 20 mM Hepes pH 7.5, containing 0.5 M NaCl. To elute a non-consumed [$^1$H + $^3$H]SAM, the columns were washed with 3 $\times$ 2 ml of 1 M $NH_4OH$. In addition, in experiments in which peptide H3N4 was the substrate of the SETD3 enzyme, fractions eluted with ammonium solution were dried out at 60°C for 48 hr to break down [$^1$H + $^3$H]SAM (*Parks and Schlenk, 1958*). The remaining residues were dissolved in 2 ml of 20 mM Hepes (pH 7.5) and refractionated on Dowex 50W-X4 columns to verify the possible presence of di- and tri-methylated forms of peptide H3N4. No such polymethylated derivatives of peptide H3N4 were detected.

In all cases, the samples to be counted were mixed with 10 ml of scintillation fluid (Ultima Gold) and the incorporated radioactivity was analyzed with a Beckman LS6000 IC liquid scintillation counter.

## Protein purification

### Purification of rat actin-specific histidine *N*-methyltransferase

A myofibrillar extract of rat skeletal muscle was prepared as described by *Vijayasarathy and Narasinga Rao (1987)*. Briefly, rat leg muscles (200 g) from 18 male WAG rats, aged 3 months, were homogenized in a Waring Blender 7011HS with three volumes (w/v) of buffer consisting of 10 mM Tris-HCl, pH 7.4, 1 mM DTT, 250 mM sucrose, 5 µg/ml leupeptin and 5 µg/ml antipain. The homogenate was centrifuged for 15 min at 10,000 $\times$ g at 4°C. The pellet was resuspended in 2400 ml of borate buffer (39 mM potassium borate, pH 7.2, 275 mM KCl, 1 mM DTT, 3 µg/ml leupeptin and 3 µg/ml antipain), extracted by shaking at 160 rpm for 1 hr at 5°C and centrifuged for 15 min at 10,000 $\times$ g at 4°C. The resulting supernatant solution (2400 ml) was then fractionated between 0% and 20% concentration (w/v) of PEG 4000. The 0–20% precipitate was dissolved in 1000 ml of buffer A (20 mM Tris-HCl, pH 8.0, 1 mM DTT, 3 µg/ml leupeptin and 3 µg/ml antipain), centrifuged for 20 min at 25,000 $\times$ g at 4°C and filtered through six layers of gauze to remove fat particles.

The clarified supernatant was applied to a DEAE-Sepharose column (350 ml bed volume) equilibrated with buffer A. The column was washed with 500 ml of buffer A and developed with a NaCl gradient (0–1 M in 962 ml) in buffer A, before fractions (6.5 ml) were collected. The enzymatically active fractions of the DEAE-Sepharose column (84 ml) were diluted to 350 ml with buffer A and applied to a Q-Sepharose column (100 ml bed volume) equilibrated with the same buffer. The column was washed with 360 ml of buffer A and the retained protein was eluted with a NaCl gradient (0–0.95 M in 500 ml in buffer A) before fractions (5 ml) were collected. The active fractions of the Q-Sepharose column (64 ml) were pooled, supplemented with solid NaCl so that a final salt concentration in the sample equal to 0.6 M was reached, and applied to a Phenyl-Sepharose six column (40 ml bed volume) equilibrated with buffer B (20 mM Tris-HCl, pH 7.2, 10 mM KCl, 1 mM DTT, 2 µg/ml leupeptin and 2 µg/ml antipain) containing 0.6 M NaCl. The hydrophobic interaction column was washed with 100 ml of buffer B with NaCl and the retained protein was eluted with a NaCl gradient (600–0 mM in 360 ml in buffer B). Fractions (5 ml) were collected. The enzymatically active fractions of the Phenyl-Sepharose column (93 ml) were pooled and loaded onto a HiScreen Blue-Sepharose column (4.7 ml bed volume) equilibrated with buffer B. The column was washed with 20 ml of buffer B and developed with a NaCl gradient (0–1.5 M in 70 ml) in buffer B, before fractions (3 ml) were collected. The active fractions of the HiScreen Blue-Sepharose column (27 ml) were pooled, supplemented with 100 mM KCl, 2 mM ATP and 0.2 mM $CaCl_2$, and concentrated to 2.5 ml in Vivaspin-20 ultrafiltration devices. Two milliliters of the ultrafiltrate were loaded onto a Superdex 200 16/60 column (120 ml bed volume) equilibrated with buffer C (20 mM Tris-HCl, pH 7.2, 100 mM KCl, 1 mM DTT, 0.2 mM ATP, 0.2 mM $CaCl_2$, 2 µg/ml leupeptin and 2 µg/ml antipain). The gel filtration column was then developed with 140 ml of buffer C and 3 ml fractions were collected. To obtain more purified enzyme preparations for a tandem mass spectrometry, the most active fraction (№ 14, 2.5 ml) from the Superdex 200 purification step was loaded onto a Reactive Red 120-Agarose column (1.5 ml bed volume) equilibrated with buffer C. The column was first washed with 6 ml of buffer C and

the retained proteins were eluted (four fractions of 3 ml) with buffer C containing an increasing concentration of NaCl (0–1.4 M).

All purification steps were performed at 4°C and the enzymatic preparation was stored at −70°C between steps.

## Overexpression and purification of the recombinant β-actin inclusion-body protein

Plasmid pCOLD I encoding human β-actin (ACTB, GenBank: NM_001101.4, pCOLD I/β-actin) was a kind gift from Dr. Minoru Tamura (Ehime University, Japan) and was prepared as described by *Tamura et al. (2011)*.

For β-actin production, *E. coli* BL21(DE3) (Agilent, USA) cells were transformed with the DNA construct and a single colony was selected to start an over-night pre-culture. 500 mL of LB broth (with 100 µg/mL ampicilin) was inoculated with 50 ml of the pre-culture and incubated at 37°C and 200 rpm until an OD600 of 0.5 was reached. The culture was placed on ice for 20 min (cold-shock) and isopropyl β-D-1-thiogalactopyranoside (IPTG) was added to a final concentration of 0.2 mM to induce protein expression. Cells were incubated for 20 hr at 15°C, 200 rpm, and harvested by centrifugation (6000 × g for 10 min). The cell paste was resuspended in 27.5 ml lysis buffer consisting of 20 mM Hepes, pH 7.5, 1 mM DTT, 1 mM ADP, 0.5 mM PMSF, 2 µg/ml leupeptin and 2 µg/ml antipain, together with 0.2 mg/ml hen egg white lysozyme (BioShop), and 1000 U Viscolase (A and A Biotechnology, Poland). The cells were lysed by freezing in liquid nitrogen and, after thawing and vortexing, the extracts were centrifuged at 4° C (20,000 × g for 30 min).

The pellet, containing inclusion bodies, was completely resuspended in buffer A (20 mM Hepes, pH 7.5, 2M urea, 0.5 M NaCl, 5 mM DTT and 2 mM EDTA) with the use of Potter-Elvehjem homogenizer and centrifuged at 4° C (20,000 × g for 10 min). The resulting pellet was then subjected to a further two rounds of sequential wash in buffer B (20 mM Hepes, pH 7.5, 0.5 M NaCl, 5 mM DTT and 2 mM EDTA) and buffer C (20 mM Hepes, pH 7.5, and 0.5 M NaCl). The washed inclusion bodies were finally solubilized in loading buffer (20 mM Tris-HCl, pH 7.5, 6 M guanidine HCl, 0.5 M NaCl, and 10 mM imidazole) and applied onto a HisTrap FF column (5 ml) equilibrated with the same buffer.

The column was washed with 20 ml of loading buffer and the bound protein was refolded by the use of a linear gradient of 6–0 M guanidine HCl in loading buffer (40 ml for 20 min). Next, the column was washed with 15 ml of loading buffer without guanidine HCl and the retained proteins were eluted with a stepwise gradient of imidazole (25 ml of 40 mM, 25 ml of 60 mM and 20 ml of 500 mM). The recombinant proteins were eluted with 500 mM imidazole in homogeneous form as confirmed by SDS-PAGE (see *Figure 1—figure supplement 1*). The β-actin preparation was immediately desalted on PD-10 columns equilibrated with 20 mM Tris-HCl, pH 7.5, 1 mM DTT, 6% sucrose, 2 µg/ml leupeptin and 2 µg/ml antipain. The purified β-actin was stored at −70°C.

## Overexpression and purification of the soluble recombinant β-actin

Plasmid pCOLD I encoding human β-actin (pCOLD I/β-actin) was used as a template for PCR amplification of β-actin cDNA. The resulting ORF was then cloned into pESC-URA yeast expression vector (pESC-URA/β-actin). *Saccharomyces cerevisiae* BY4742 strain (EUROSCARF, Germany) was transformed with the above-mentioned plasmid using the high-efficiency lithium acetate transformation method (*Gietz and Woods, 2002*). The resulting yeast strain was designated as BY4742/β-actin.

The method for the purification of recombinant β-actin expressed in *S. cerevisiae* was that reported by *Karlsson (1988)*. BY4742/β-actin was cultivated in 500 ml of SG medium (0.67% yeast nitrogen base without amino acids, 2% galactose, supplemented with histidine, leucine and lysine) for 24 hr. Cells were collected, washed with water, and suspended in 30 ml of buffer C (5 mM Tris-HCl pH 7.6, 0.2 mM CaCl$_2$, 0.2 mM DTT, and 0.3 mM PMSF). After the addition of 100 U of DNase I, cells were homogenized in Braun MSK cell homogenizer using glass beads (B. Braun Melsungen). Crude lysate was centrifuged (5000 × g, 5 min), diluted twice with the buffer, and applied to the Ni-NTA-agarose column (Sigma-Aldrich). Next, the column was washed with 30 ml of buffer C supplemented with 8 mM imidazole, and retained proteins were eluted with buffer C containing 500 mM imidazole (≈15 ml, 0.7 ml fractions). Protein concentration in each fraction was estimated by SDS-PAGE and the most concentrated samples were taken for further procedures.

To obtain a yeast-produced refolded human β-actin, pellet of yeast cells (1000 ml culture) was suspended in 30 ml of buffer D (20 mM Tris-HCl pH 7.2, 0.5 M NaCl, 1 mM DTT, and 100 U of DNase I) and homogenized in a Braun MSK cell homogenizer. Crude lysate was then supplemented with 6 M guanidine HCl and 10 mM imidazole, centrifuged (20,000 × g, 20 min) and applied on a HisTrap FF column (5 ml). The subsequent on-column refolding and purification of β-actin was performed as described for *E. coli*-derived protein.

For preparation of β-actin complexes with either cofilin or profilin (see below), an equimolar amount of the recombinant homogenous actin-binding protein was mixed with the yeast-produced β-actin and dialyzed twice against 1000 ml of dialysis buffer (20 mM Tris-HCl pH 7.6, 1 mM DTT, 6% sucrose, and 0.2 mM CaCl$_2$), for 2 hr in 250 ml and then overnight in 500 ml. When obtaining β-actin complexed with either ADP or ATP, samples eluted from Ni-NTA-agarose were dialyzed under the same conditions against dialysis buffer containing 50 µM of an appropriate nucleotide. In all cases, dialyzed protein preparation was concentrated using an Amicon 10K device (Merck, Germany).

## Overexpression in COS-7 cells and purification of the recombinant SETD3 proteins

Rat total RNA was prepared from 200 mg of leg muscles with the use of TriPure reagent according to the manufacturer's instructions, whereas human skeletal muscle total RNA was purchased from Clontech (USA). cDNA was synthesized using Moloney murine leukemia virus-reverse transcriptase (Thermo-Fermentas), with oligo(dT)$_{18}$ primer and 2.5 µg total RNA according to the manufacturer's instructions.

The open reading frames encoding rat (NCBI Reference Sequence: XM_002726774.2) and human (NM_032233.2) SETD3 protein were PCR-amplified using Pfu DNA polymerase in the presence of 1 M betaine. The SETD3 genes were amplified using 5′ primers containing the initiator codon preceded by the Kozak consensus sequence (*Kozak, 1987*) and an KpnI site, and 3′ primers in which the original stop codon was replaced by an amino-acid-coding codon flanked by a NotI site (for primer sequences, see *Table 5*). The amplified DNA products of the expected size were digested with the appropriate restriction enzymes, cloned in the pEF6/Myc-His A expression vector (Invitrogen, USA) (which allows the production of proteins with an C-terminal His$_6$-tag), and verified by

**Table 5.** Sequences of primers used in PCR experiments The nucleotides corresponding to the coding sequences are in capital letters, the Kozak consensus sequence is shown in bold, and the added restriction sites are underlined.

| Primer | Sequence | Restriction site | Plasmid | Protein expressed |
|---|---|---|---|---|
| | *Preparation of SEDT3 protein expression vectors* | | | |
| #1 | taaggtacc**gccacc**ATGGGTAAGAAGAGTCGAGTG | KpnI | pEF6/Myc-His A | C-terminal His$_6$-tagged rat SEDT3 protein |
| #2* | taagcggccgCCAGAGTCGCTCCTTCACCGC | NotI | | |
| #3 | taaggtacc**gccacc**ATGGGTAAGAAGAGTCGAGTAA | KpnI | pEF6/Myc-His A | C-terminal His$_6$-tagged rat SEDT3 protein |
| #4* | taagcggccgCCACTCCTTAACTCCAGCAGTG | NotI | | |
| | *Preparation of cofilin-1 and profilin-1 expression vectors* | | | |
| #5 | tatacatATGGCCTCCGGTGTGGCTG | NdeI | pCOLD I | N-terminal His$_6$-tagged human cofilin-1 |
| #6 | tataaagcTTATCACAAAGGCTTGCCCTCCAG | HindIII | | |
| #7 | tatacatATGGCCGGGTGGAACGCC | NdeI | pCOLD I | N-terminal His$_6$-tagged human profilin-1 |
| #8 | tataaagcTTATCAGTACTGGGAACGCCGAAG | HindIII | | |
| | *Preparation of S-adenosyl-L-homocysteine nucleosidase and andenine deaminase expression vectors* | | | |
| #9 | tatacatATGAAAATCGGCATCATTGGTG | NdeI | pCOLD I | N-terminal His$_6$-tagged *E. coli* SAH nucleosidase |
| #10 | tataaagcTTAGCCATGTGCAAGTTTCTGC | HindIII | | |
| #11 | tataggtaccTTGAATAAAGAAGCGCTAGTCAAT | KpnI | pCOLD I | N-terminal His$_6$-tagged *B. subtilis* andenine deaminase |
| #12 | tataggatccTTATTGCAGTGATATGTGTTGAAAT | BamHI | | |

*Original STOP codons were replaced by CCA (shown in italic and underlined) to allow C-terminal HisTag translation.

DOI: https://doi.org/10.7554/eLife.37921.044

DNA sequencing (Macrogen, The Netherlands). The resulting vector was designated as pEF6/SETD3. For transfections, COS-7 cells (Cell Lines Service, Germany) were plated in 100 mm Petri dishes at a cell density of $1.7 \times 10^6$ or $2.1 \times 10^6$ cells per plate in Dulbecco's minimal essential medium supplemented with 100 units/ml penicillin, 100 µg/ml streptomycin, and 10% (v/v) fetal bovine serum, and grown in a humidified incubator under 95% air and 5% $CO_2$ atmosphere at 37°C. After 24 hr, each plate was transfected with 7–8 µg of either unmodified pEF6/Myc-His A vector or pEF6/SETD3 using the TurboFect transfection reagent according to the protocol provided by the manufacturer. After 48 hr, the culture medium was removed, the cells were washed with 5 ml phosphate buffered saline and harvested in 1 ml of 20 mM Tris-HCl pH 7.2, containing 10 mM KCl, 1 mM DTT, 5 µg/ml leupeptin and 5 µg/ml antipain. The cells were lysed by freezing in liquid nitrogen and, after thawing and vortexing, the extracts were centrifuged at 4° C (20,000 × g for 30 min) to remove insoluble material.

For the purification of recombinant SETD3 proteins, the supernatant of the COS-7 lysate (13–18 ml) was diluted 3-fold with buffer A (50 mM Tris-HCl pH 7.4, 400 mM NaCl, 10 mM KCl, 30 mM imidazole, 3 µg/ml leupeptin and 3 µg/ml antipain) and applied onto a HisTrap HP column (1 ml) equilibrated with the same buffer. The column was washed with 6–10 ml buffer A and the retained protein was eluted with a stepwise gradient of imidazole (6 ml of 60 mM, 6 ml of 150 mM and 6 ml of 300 mM) in buffer A. The recombinant proteins were eluted with 150 mM imidazole in homogeneous form, as confirmed by SDS-PAGE (see *Figure 6—figure supplement 1*). The enzyme preparations were desalted on PD-10 columns equilibrated with 20 mM Tris-HCl pH 7.2, 50 mM KCl, 1 mM DTT, 6% sucrose, 2 µg/ml leupeptin and 2 µg/ml antipain. The yield of recombinant proteins was 0.5–0.6 mg of homogenous protein per about 25–40 mg of soluble COS-7 cell protein. The purified enzymes were aliquoted and stored at −70°C. Repeated freeze-thawing was avoided because it impairs SETD3 activity.

## Overexpression in *E. coli* and purification of the recombinant SETD3 proteins

Bacterial expression plasmids for human and rat SETD3 fused to an N-terminal His$_6$-tag were constructed by sub-cloning the open reading frames from the pEF6/SETD3 plasmids into the pCOLD I vector (Takara Bio, Kusatsu, Japan), using KpnI and XbaI enzymes (pCOLD I/SETD3).

For protein production, *E. coli* BL21(DE3) cells were transformed with an appropriate DNA construct and a single colony was selected to start an over-night pre-culture. 500 mL of LB broth (with 100 µg/mL ampicilin) was inoculated with 50 ml of the pre-culture and incubated at 37°C and 200 rpm until an OD600 of 0.6 was reached. The culture was placed on ice for 20 min (cold-shock) and IPTG was added to a final concentration of 0.3 mM to induce protein expression. Cells were incubated for 16 hr at 13°C, 200 rpm, and harvested by centrifugation (6000 × g for 10 min). The cell paste was resuspended in 25 ml lysis buffer consisting of 25 mM Hepes pH 7.5, 300 mM NaCl, 10 mM KCl, 1 mM DTT, 2 mM $MgCl_2$, 1 mM PMSF, 0.25 mg/ml hen egg white lysozyme (BioShop, Canada) and 1250 U Viscolase (A and A Biotechnology, Poland). The cells were lysed by freezing in liquid nitrogen and, after thawing and vortexing, the extracts were centrifuged at 4° C (20000 × g for 20 min).

For the purification of recombinant SETD3 proteins, the supernatant of *E. coli* lysate (22 ml) was diluted 3-fold with buffer A (50 mM Hepes pH 7.5, 300 mM NaCl, 10 mM KCl, 30 mM imidazole, 1 mM DTT and 0.5 mM PMSF) and applied onto a HisTrap FF column (5 ml) equilibrated with the same buffer. The column was then washed with 35 ml buffer A, and the retained proteins were eluted with a stepwise gradient of imidazole (15 ml of 60 mM, 16 ml of 150 mM and 16 ml of 300 mM) in buffer A. The recombinant proteins were present in both 150 mM and 300 mM imidazole fractions (see *Figure 7—figure supplement 1*), but only SETD3 protein eluted at the highest concentration of imidazole, exhibiting >95% purity as confirmed by SDS-PAGE, was further processed. The enzyme preparation was desalted onto PD-10 columns equilibrated with 20 mM Tris-HCl pH 7.2, 50 mM KCl, 1 mM DTT and 6% sucrose. The yield of recombinant proteins was 6.6 mg and 8.2 mg of homogenous rSETD3 and hSETD3, respectively, per 500 ml of culture. The purified proteins were aliquoted and stored at −70°C.

## Overexpression and purification of the recombinant cofilin and profilin proteins

Human skeletal muscle total RNA was obtained from Clontech (USA) and cDNA was synthesized as described above. The open reading frames encoding human cofilin-1 and profilin-1 (NCBI Reference Sequence: NM_005507.2 and NM_005022.3, respectively) were PCR-amplified using Pfu DNA polymerase in the presence of 1 M betaine. ORFs of both proteins were amplified using 5′ primers containing the initiator codon preceded by an NdeI site and 3′ primers flanked by a HindIII site (for primer sequences, see *Table 5*). The amplified DNA products of the expected size were digested with the appropriate restriction enzymes and cloned into the pCOLD I expression vector (Takara Bio, Kusatsu, Japan), which allows the production of proteins with an N-terminal His$_6$-tag (pCOLD I/Cofilin-1 and pCOLD I/Profilin-1), and verified by DNA sequencing (Macrogen, The Netherlands).

For protein production, *E. coli* BL21(DE3) cells were transformed with the appropriate DNA construct, and a single colony was selected to start an over-night pre-culture. 200 mL of LB broth (with 100 µg/mL ampicilin) was inoculated with 20 ml of the pre-culture and incubated at 37°C and 200 rpm until an OD600 of 0.6 was reached. The culture was placed on ice for 20 min (cold-shock) and IPTG was added to a final concentration of 0.25 mM to induce protein expression. Cells were incubated for 16 hr at 13°C, 200 rpm, and harvested by centrifugation (6000 × g for 10 min). The cell paste was resuspended in 10 ml lysis buffer consisting of 25 mM Hepes pH 7.5, 300 mM NaCl, 50 mM KCl, 1 mM DTT, 2 mM MgCl$_2$, 1 mM PMSF, 5 µg/ml leupeptin and 5 µg/ml antipain, together with 0.25 mg/ml hen egg white lysozyme (BioShop, Canada) and 50 U DNase I (Roche). The cells were lysed by freezing in liquid nitrogen and, after thawing and vortexing, the extracts were centrifuged at 4° C (20,000 × g for 30 min).

For the purification of recombinant actin-binding proteins, the supernatant of *E. coli* lysate (10 ml) was diluted 4-fold with buffer A (50 mM Hepes pH 7.5, 300 mM NaCl, 10 mM KCl, 30 mM imidazole and 1 mM DTT) and applied onto a HisTrap FF column (5 ml) equilibrated with the same buffer. The column was sequentially washed with 23 ml buffer A and 23 ml of the same buffer supplemented with urea at 1.8 M concentration. Next, the retained protein was eluted with a stepwise gradient of imidazole (23 ml of 60 mM, 23 ml of 150 mM and 22 ml of 300 mM) in buffer A. The recombinant proteins were eluted with 300 mM imidazole in homogeneous form, as confirmed by SDS-PAGE (not shown). The enzyme preparations were dialyzed three times against 400 ml of dialysis buffer consisting of 20 mM Tris-HCl pH 7.5, 200 mM NaCl, 1 mM DTT and 6% sucrose. The yield of recombinant proteins was 7.1 and 17.7 mg of homogenous cofilin-1 and profilin-1, respectively, per 200 ml of *E. coli* culture. The purified proteins were aliquoted and stored at −70°C.

## Overexpression and purification of the recombinant SAH nucleosidase and adenine deaminase

*E. coli* DNA was extracted by heating 50 µl of over-night cultured BL21(DE3) cells at 95°C for 15 min, whereas *Bacillus subtilis* (ATCC 6633, Sigma-Aldrich) genomic DNA was purified from 100 mg of bacterial cells with the use of TriPure reagent according to the manufacturer's instructions.

The open reading frames encoding *E. coli* SAH nucleosidase and *B. subtilis* adenine deaminase (NCBI Reference Sequence: NC_000913.3 and NC_000964.3, respectively) were PCR-amplified using either Pfu DNA polymerase alone or a mixture of Taq:Pfu polymerases (1:0.2), respectively, in the presence of 1 M betaine. SAH nucleosidase ORF was amplified using a 5′ primer containing the initiator codon preceded by an NdeI site and a 3′ primer with a HindIII site, whereas adenine deaminase DNA was amplified using a 5′ primer with the initiator codon preceded by an KpnI site and a 3′ primer with a BamHI site (for primer sequences, see *Table 5*). The amplified DNA products of expected size were digested with the appropriate restriction enzymes, cloned into the pCOLD I expression vector (pCOLD I/SAH nucleosidase and pCOLD I/adenine deaminase) and verified by DNA sequencing (Macrogen, The Netherlands).

For protein production, *E. coli* BL21(DE3) cells were transformed with the appropriate DNA construct and a single colony was selected to start an over-night pre-culture. 100 mL of LB broth (with 100 µg/mL ampicilin) was inoculated with 10 ml of the pre-culture and incubated at 37°C and 200 rpm until an OD600 of 0.6 was reached. The culture was placed on ice for 20 min (cold-shock) and IPTG was added to a final concentration of 0.25 mM to induce protein expression. Cells were incubated for 16 hr at 13°C, 200 rpm, and harvested by centrifugation (6000 × g for 10 min). The cell

pellet was resuspended in 10 ml lysis buffer consisting of 25 mM Hepes pH 7.5, 300 mM NaCl, 50 mM KCl, 1 mM DTT, 2 mM MgCl$_2$, 1 mM PMSF, 5 µg/ml leupeptin and 5 µg/ml antipain, together with 0.25 mg/ml hen egg white lysozyme (BioShop) and 25 U DNase I (Roche). The cells were lysed by freezing in liquid nitrogen and, after thawing and vortexing, the extracts were centrifuged at 4°C (20,000 × g for 30 min).

For protein purification, the supernatant of the *E. coli* lysate (10 ml) was diluted 3-fold with buffer A (50 mM Tris-HCl, pH 7.2, 400 mM NaCl, 10 mM KCl, 30 mM imidazole, 1 mM DTT, 3 µg/ml leupeptin and 3 µg/ml antipain) and applied onto a HisTrap FF column (5 ml) equilibrated with the same buffer. The column was washed with 20–30 ml buffer A and the retained protein was eluted with a stepwise gradient of imidazole (25 ml of 60 mM, 20 ml of 150 mM and 20 ml of 300 mM) in buffer A. The recombinant proteins were eluted with 150–300 mM imidazole in homogeneous form, as confirmed by SDS-PAGE (not shown). The enzyme preparations were desalted onto PD-10 columns equilibrated with 20 mM Tris-HCl pH 7.2, 50 mM KCl, 1 mM DTT, 6% sucrose, 2 µg/ml leupeptin and 2 µg/ml antipain. The yield of recombinant proteins was 1.2 mg and 3.1 mg of homogenous adenine deaminase and SAH nucleosidase, respectively, per 200 ml of *E. coli* culture. The purified enzymes were aliquoted and stored at −70°C.

## Identification of the rat actin-specific histidine *N*-methyltransferase by tandem mass spectrometry

A preliminary SDS-PAGE analysis of the peak activity fractions from the Reactive Red 120-agarose purification step revealed the presence of very faint protein bands with a low protein content, consequently the fractions were 15-fold concentrated in a Vivaspin-500 ultrafiltration device and reanalyzed by SDS-PAGE. All bands present in the concentrated fractions were cut from a 10% gel and digested with trypsin. In-gel digestions of the peptides were performed as described (*Shevchenko et al., 2006*). Peptides were analyzed by nanoUPLC-tandem mass spectrometry employing Acquity nanoUPLC coupled with a Synapt G2 HDMS Q-TOF mass spectrometer (Waters, Milford, USA) fitted with a nanospray source and working in MSE mode under default parameters. Briefly, the products of in-gel protein digestion were loaded onto a Waters Symmetry C18 trapping column (20 mm × 180 µm) coupled to the Waters BEH130 C18 UPLC column (250 mm × 75 µm). The peptides were eluted from these columns in a 1–85% gradient of acetonitrile in water (both containing 0.1% formic acid) at a flow rate of 0.3 µl/min. The peptides were directly eluted into the mass spectrometer. Data were acquired and analyzed using MassLynx 4.1 software (Waters, USA) and ProteinLynx Global Server 2.4 software (PLGS, Waters, USA) with a False Discovery Rate ≤ 4%. To identify actin-specific histidine *N*-methyltransferase, the complete rat (*Rattus norvegicus*) reference proteome was downloaded from the NCBI Protein database, randomized and used as a databank for the MS/MS software.

## Generation of the HAP1 cell lines deficient in SETD3 by the CRISPR/Cas9 method

Human-derived HAP1 cells (Horizon Discovery, Waterbeach, UK) have a haploid karyotype except for a disomy of chromosome 8 (25, XY,+8, Ph+) and are therefore considered near-haploid. The cell line expresses *Setd3* gene at the level of 32.04 transcripts per kilobase million (https://www.horizon-discovery.com/human-setd3-knockout-cell-line-hzghc84193). Since *Setd3* gene is on chromosome 14, all generated *Setd3*-defficent HAP1 cells will show the complete loss of SETD3 expression. The parental cells were therefore used to create HAP1 cell lines that are deficient in SETD3 by using CRISPR/Cas9 technology.

Two CRISPR/Cas9 constructs were generated to target exon 4 of the human *Setd3* gene by ligating two different sets of annealed primer pairs (CRISPR-hSETD3-s1 CACCGgttctgtcgagggttttgaaa, CRISPR-hSETD3-as1 AAACtttcaaaaccctcgacagaacC and CRISPR-hSETD3-s2 CACCGgaggcccatttcattagatc, CRISPR-hSETD3-as2 AAACgatctaatgaaatgggcctcC). The annealed primer pairs were ligated into the vector pSpCas9n(BB)−2A-Puro (PX462) V2.0 (a gift from F. Zhang, Massachusetts Institute of Technology; Addgene plasmid no. 62987) (*Ran et al., 2013*) digested by BbsI. Constructs were validated by sequencing (Beckman Coulter Genomics). HAP1 cells were cultured in IMDM (Iscove's Modified Dulbecco's Medium) containing 10% FBS, 2 mM L-glutamine and penicillin/streptomycin (Thermo-Life Technologies). Cells were transfected with the CRISPR constructs exactly as described

in *Collard et al., 2016*. Genomic DNA from puromycin-resistant clones was used to PCR-amplify the regions encompassing the targeted site and the PCR products were sequenced to evaluate the gene modification present in each clone. The three SETD3-KO clones that were ultimately selected for further studies presented the following modifications: clone KO-A1, a 16 bp deletion after Phe71, which introduces a frame shift and a premature stop codon at position 75; clone KO-A3, a 4 bp deletion and 6 bp insertion that introduce a frame shift after Met83 and a premature stop codon at position 109; and clone KO-A5, a 18 bp deletion and 107 bp insertion after Trp85, with a premature stop codon at position 292.

The absence of SETD3 protein from each SETD3-KO cell lines was confirmed by western blotting performed as described previously (*Drozak et al., 2013*), with the use of rabbit polyclonal primary antibody against human SETD3 (ab174662, Abcam, UK) and a horseradish peroxidase-conjugated goat anti-rabbit secondary IgG antibody (AS09602, Agrisera, Vannas, Sweden).

## Product analysis

### Detection of in vitro methylated β-actin by tandem mass spectrometry

To obtain a sufficient amount of the methylated β-actin formed in the reaction catalyzed by recombinant rat or human SETD3 protein for mass spectrometry analysis, the reaction mixture was scaled up. Briefly, 15 µg recombinant human β-actin was incubated for 90 min at 37°C in 70 µl of a reaction mixture containing 25 mM Tris-HCl pH 7.2, 10 mM KCl, 1 mM DTT, 20 µM $MnCl_2$, 6% sucrose, 20–30 µM [$^2$H]SAM, 4.5–9.0 µg SAH nucleosidase and 1.75–3.5 µg adenine deaminase in the absence or presence of 1.1–3.5 µg SETD3 protein. The incubation was stopped by the addition of 30 µl of the reaction mixture to 10 µl 4 × Laemmli buffer and by heating the sample for 5 min at 95°C, while the remaining 40 µl of the mixture was flesh-frozen in $LN_2$ and stored at −70°C. The yield of β-actin methylation was about 30%, as determined for the parallel labeling reaction in the presence of [$^2$H +$^3$H]SAM.

For MS/MS analysis of deuterated protein, 35 µl of denatured sample (≈18 µg protein) was separated in 10% SDS-PAGE and silver stained (*Shevchenko et al., 1996*). The bands corresponding to β-actin were excised from the gels, destained and digested with trypsin according to the methods described in *Shevchenko et al., 2006*. In addition, 18 µl of the flesh-frozen sample (12 µg protein) was trypsin-digested 'in solution' for 16 hr at 30°C in 30 µl of the reaction mixture containing 100 mM $NH_4HCO_3$ and 1.5 µg trypsin (MS grade, Sigma-Aldrich). The digestion reaction was terminated by the addition of trifluoroacetic acid to a final concentration of 1%. Resulting peptides were analyzed by nanoUPLC-tandem mass spectrometry, as described above in the section 'Identification of rat actin-specific histidine N-methyltransferase by tandem mass spectrometry'. To identify actin peptides, the full amino-acid sequence of recombinant human β-actin in fusion with the His$_6$-tag at the N-terminus was used as a databank for the MS/MS software. Detection of the trideuterium-methylated peptides was performed in a fully automatic mode by PLGS 2.4 software (Waters, USA), following its update by adding trideuterium-methylation (+17.03448 Da) of Cys, Asp, Asn, His, Lys, Arg, Glu and Gln residues as a possible modification of peptides.

### Detection of methylated β-actin from HAP1 human cell lines by tandem mass spectrometry

HAP1 wildtype (WT-A1 and WT-A2) and SETD3-KO cells (KO-A1 and KO-A3) were cultured in IMDM containing 10% FBS, 2 mM L-glutamine and penicillin/streptomycin (Life Technologies), each in four 10 cm diameter plates. At confluence, the medium was rapidly removed, the plates were washed with 5 ml cold PBS and each of them scraped in 0.8 ml of buffer A (25 mM Hepes pH 7.4, 20 mM NaCl, 2.5 µg/ml leupeptin and 2.5 µg/ml antipain) and combined to collect the cells. The four soluble cell extracts were prepared by two cycles of freezing the cell suspension in liquid nitrogen and thawing before vortex-mixing and centrifuging for 15 min at 15,000 × g and 4°C to recover soluble proteins.

To purify the actin partially, the cell lysates were diluted 5-fold in buffer A (20 ml final volume) and loaded onto a 1 ml-HiTrap Q HP (GE Healthcare) equilibrated with the same buffer. The column was washed with 6 ml of buffer A and proteins were eluted with a linear NaCl gradient (20–750 mM in 20 ml buffer A). The elution fractions (1 ml) were analysed by SDS-PAGE followed by Coomassie

blue staining and by western blotting using a rabbit anti-actin antibody (A2066, Sigma-Aldrich). The actin band of the richest fraction was cut out of the gel and digested with trypsin.

The resulting peptides were analysed by LC-MS/MS with a LTQ XL (ThermoFisher Scientific) equipped with a microflow ESI source interfaced with a Dionex Ultimate Plus Dual gradient pump, a Switchos column switching device, and a Famos Autosampler (Dionex, Amsterdam, The Netherlands) as described (*Houddane et al., 2017*). Raw data files were then analyzed by bioinformatics using Proteome Discoverer (Thermo Fisher Scientific). Peak lists were generated using the application spectrum selector in the Proteome Discoverer 1.4 package. From raw files, MS/MS spectra were exported as individual files in the .dta format with the following settings: peptide mass range, 400–3500 Da; minimal total ion intensity, 500; and minimal number of fragment ions, 12. The resulting peak lists were searched using SequestHT against a human protein database obtained from Uniprot and containing only actin sequences. The following parameters were used: trypsin was selected with semi-proteolytic cleavage only after lysine and arginine; the number of internal cleavage sites was set to 2; the mass tolerance for precursor and fragment ions was 1.0 Da; and the considered dynamic modifications were + 15.99 Da for oxidized methionine,+14.00 Da for methylation on histidine and + 42.00 Da for acetylation of N-terminus or lysines. Peptide spectral matches (PSM) were filtered using charge-state versus cross-correlation scores (Xcorr) and methylation sites were validated manually.

## Detection of actin methylation in *Drosophila melanogaster* by tandem mass spectrometry

To investigate whether the absence of SETD3 in *D. melanogaster* also prevented actin methylation, we prepared protein extracts by homogenizing approximately 20 larvae from wildtype or SETD3 KO flies (*Tiebe et al., 2018*) in 0.5 ml of 25 mM Hepes pH 7.5 with 2.5 µg/ml of leupeptin and antipain. After centrifugation (15 min at 16 000 ×g), the supernatant was recovered, protein concentration was measured (*Bradford, 1976*) and 10 µg of total proteins were digested with trypsin and used for LC-MS/MS analysis. Peptides were dissolved in solvent A (0.1% trifluoroacetic acid (TFA) in 2% acetonitrile (ACN)), directly loaded onto reversed-phase pre-column (Acclaim PepMap 100, Thermo Scientific) and eluted in backflush mode. Peptide (250 ng) separation was performed using a reversed-phase analytical column (Acclaim PepMap RSLC, 0.075 × 250 mm, Thermo Scientific) with a linear gradient of 4–27.5% solvent B (0.1% formic acid (FA) in 98% ACN) for 100 min, 27.5–40% solvent B for 10 min, 40–95% solvent B for 1 min and holding at 95% for the last 10 min at a constant flow rate of 300 nl/min on an EASY-nLC 1000 ultra-performance liquid chromatography (UPLC) system. The peptides were analyzed by an Orbitrap Fusion Lumos tribrid mass spectrometer (ThermoFisher Scientific). The peptides were subjected to NSI source followed by tandem mass spectrometry (MS/MS) in Fusion Lumos coupled online to the UPLC. Intact peptides were detected in the Orbitrap at a resolution of 120,000. Peptides were selected for MS/MS using a higher-energy C-trap dissociation (HCD) setting of 30; ion fragments were detected in the ion trap. A data-dependent procedure that alternated between one MS scan followed by 20 MS/MS scans was applied for the top 20 precursor ions above a threshold ion count of 5.0E3 in the MS survey scan with 30.0 s dynamic exclusion. The electrospray voltage applied was 2.1 kV. MS1 spectra were obtained with an automatic gain control (AGC) target of 4E5 ions and a maximum injection time of 50 ms, whereas MS2 spectra were acquired with an AGC target of 5E3 ions and a maximum injection time of 300 ms. For MS scans, the m/z scan range was 350 to 1500. The resulting MS/MS data were processed using the Sequest HT search engine within Proteome Discoverer 2.2 against a protein database containing *Drosophila melanogaster* sequences obtained from Swissprot (4810 entries, version 27 10 2017). Trypsin was specified as cleavage enzyme allowing up to two missed cleavages, four modifications per peptide and up to five charges. Mass error was set to 10 ppm for precursor ions and 0.6 Da for fragment ions. Oxidation on Met, carbamidomethyl on Cys and methylation on His (+14.015 Da) were considered as variable modifications. False discovery rate (FDR) was assessed using Percolator and thresholds for protein, peptide and modification site were specified at 1%.

## Phenotypic analysis

### Glucose consumption and lactate production

HAP1 cells ($150 \times 10^3$) were seeded in 12-well plates and incubated in IMDM with 10% FBS for 40 hr with a change of medium every 20 hr. Following 40 hr incubation, cell confluency was about 90% and the fresh medium was added to start the final incubation lasting for the next 20 hr. To finish the incubation, 1 ml of the culture medium was transferred to 100 µl of 35% $HClO_4$ and centrifuged at $10,000 \times$ g for 10 min, before the resulting supernatant was neutralized with 3 M $K_2CO_3$/3 M KOH. The lactate produced was determined spectrophotometrically using lactate dehydrogenase (Roche) (**Noll, 1984**), whereas glucose consumption was measured in a spectrophotometric assay that uses hexokinase and glucose-6-phosphate dehydrogenase (Roche) (**Kunst and Draeger, 1984**). Adherent cells were washed three times with 3 ml of PBS and then solubilized in 0.1 M NaOH containing 0.125% Triton X-100 for protein quantification (Bradford) to allow normalization of lactate produced and glucose consumed to biomass.

## Organization of actin cytoskeleton

Visualization of actin filaments was performed as described Domanski et al., 2016**Domanski et al., 2016**. Briefly, HAP1 cells, both control and *Setd*3-KO clones, were cultured in IMDM supplemented with 10% FBS, penicillin (100 U/ml) and streptomycin (100 µg/ml) at 37°C in a humidified atmosphere containing 5% $CO_2$. For confocal microscopy, $5 \times 10^5$ cells were seeded on a $22 \times 22$ mm glass coverslip in a 35 mm plate. Either 24 or 48 hr after cell seeding, culture medium was removed and the cells were washed three times with PBS, fixed with 3% paraformaldehyde for 10 min at room temperature, permeabilized with 0.05% Triton X-100 in PBS for 3 min and washed again three times with PBS. F-actin was visualized by staining with TRITC-phalloidin (Sigma-Aldrich) in PBS for 1 hr at room temperature. Nuclei were localized with Hoechst 33342 dye (Sigma-Aldrich). After rinsing in PBS, coverslips were mounted using fluorescence mounting medium (DakoCytomation-Agilent, Santa Clara, USA). Samples were analyzed with the use of a Zeiss Axio Observer.Z1 LSM 700 confocal laser scanning microscope equipped with a Plan-Apochromat 63/1.40 Oil objective (Zeiss, Oberkochen, Germany). To detect the fluorescence of TRITC-phalloidin, a 555 nm excitation line was used, and for detecting blue fluorescence of Hoechst 33342, a 405 nm excitation line was employed. For 3D analysis, confocal Z-stacks comprising 10–13 optical slices were reconstructed into a 3D image using Zeiss Zen software with default settings.

## Analysis of the impact of actin methylation on its thermal stability

For determination of the thermostability of actin protein present in cell-free lysates of HAP1 wildtype (WT-A2) and SETD3-KO cells (KO-A3), 40 µl of lysates (73.6 µg protein) were incubated for 5 min at 4, 30, 40, 43, 46, 49, 52, 55 and 60°C, followed by a 5 min incubation at 4°C. The denatured proteins were removed in the pellet following centrifugation ($13,000 \times$ g, 10 min) and the remaining non-denatured and still soluble proteins present in the supernatant were subsequently separated by SDS-PAGE. Soluble actin was detected by western blotting with the use of rabbit polyclonal primary antibody against vertebrate actin (A2066, Sigma-Aldrich, USA) and a horseradish peroxidase-conjugated goat anti-rabbit secondary IgG antibody (AS09602, Agrisera, Sweden). The quantitative analysis of the protein band corresponding to actin was performed by densitometry using Quantity One software (BioRad, Hercules, USA).

## Phylogenetic analysis

Sequences homologous to rat SETD3 protein (NCBI Reference Sequence: XM_002726774.2) were identified by Protein Blast searches. A phylogenetic analysis was performed on the Phylogeny.fr platform (www.phylogeny.fr) (**Dereeper et al., 2008**). Amino acid sequences were aligned with MUSCLE (v3.7) (**Edgar, 2004**). After alignment, ambiguous regions were removed with Gblocks (v0.91b) (**Castresana, 2000**). Phylogenetic trees were generated using the Phylogenetic estimation using Maximum Likelihood (PhyML) (**Guindon and Gascuel, 2003**) with the WAG model for amino acid substitution (**Whelan and Goldman, 2001**). The final tree was customized with the editing interface TreeDyn (**Chevenet et al., 2006**). A confidence level was assessed using the approximate likelihood ratio test (aLRT) (minimum of SH-like and $\chi^2$-based parametric) (**Anisimova and Gascuel, 2006**).

## Analytical methods

Protein concentration was determined spectrophotometrically according to *Bradford (1976)* using bovine γ-globulin as a standard. When appropriate, the $His_6$-tagged recombinant proteins were detected by western blot analysis, employing a mouse primary antibody against $His_6$-tag (27-4710-01, GE Healthcare, USA) and a horseradish peroxidase-conjugated goat anti-mouse antibody (A2554, Sigma-Aldrich, USA), as described previously (*Drozak et al., 2013*). All western-blotting analyses employed chemiluminescence and signals acquisition with Amersham Hyperfilm ECL, with the pattern of the prestained protein ladder being copied from the blotting membrane onto the film using a set of felt-tip pens.

## Calculations

$V_{max}$, $K_M$ and $k_{cat}$ for the methyltransferase activities of the studied enzymes were calculated with Prism 4.0 (GraphPad Software, La Jolla, USA) using a nonlinear regression.

## Acknowledgments

The authors would like to express their appreciation to Professor Jadwiga Bryla (University of Warsaw) who read the first draft of this article and provided numerous helpful suggestions.

## Additional information

### Funding

| Funder | Grant reference number | Author |
| --- | --- | --- |
| Narodowe Centrum Nauki | Opus Grant (DEC-2013/11/B/NZ1/00078) | Jakub Drozak |
| Fonds De La Recherche Scientifique - FNRS | Chercheurs qualifié | Maria Veiga-da-Cunha |
| European Commission | The Erasmus+ Programme | Marianna Terreri |
| European Commission | CePT infrastructure | Adam K Jagielski Jakub Drozak |
| Fonds De La Recherche Scientifique - FNRS | | Didier Vertommen |
| Ministerstwo Nauki i Szkolnictwa Wyższego | DSM 501-D114-86-011500-23 | Sebastian Kwiatkowski |

The funders had no role in study design, data collection and interpretation, or the decision to submit the work for publication.

### Author contributions

Sebastian Kwiatkowski, Data curation, Funding acquisition, Validation, Investigation, Writing—review and editing; Agnieszka K Seliga, Data curation, Validation, Investigation, Writing—review and editing; Didier Vertommen, Resources, Data curation, Funding acquisition, Validation, Investigation, Methodology, Writing—review and editing; Marianna Terreri, Funding acquisition, Investigation, Writing—review and editing; Takao Ishikawa, Adam K Jagielski, Resources, Investigation, Writing—review and editing; Iwona Grabowska, Resources, Data curation, Validation, Investigation, Methodology, Writing—review and editing; Marcel Tiebe, Aurelio A Teleman, Resources, Writing—review and editing; Maria Veiga-da-Cunha, Conceptualization, Resources, Data curation, Formal analysis, Supervision, Funding acquisition, Validation, Investigation, Methodology, Writing—review and editing; Jakub Drozak, Conceptualization, Resources, Data curation, Formal analysis, Supervision, Funding acquisition, Validation, Investigation, Visualization, Methodology, Writing—original draft, Project administration, Writing—review and editing

## Author ORCIDs

Sebastian Kwiatkowski iD http://orcid.org/0000-0003-4908-1633
Takao Ishikawa iD http://orcid.org/0000-0002-3558-0880
Iwona Grabowska iD http://orcid.org/0000-0001-6455-9434
Marcel Tiebe iD http://orcid.org/0000-0002-7699-0587
Aurelio A Teleman iD http://orcid.org/0000-0002-4237-9368
Maria Veiga-da-Cunha iD http://orcid.org/0000-0002-2968-7374
Jakub Drozak iD http://orcid.org/0000-0002-3601-3845

## Decision letter and Author response

Decision letter https://doi.org/10.7554/eLife.37921.047
Author response https://doi.org/10.7554/eLife.37921.048

## Additional files

### Supplementary files

• Transparent reporting form
DOI: https://doi.org/10.7554/eLife.37921.045

### Data availability

All data generated or analysed during this study are included in the manuscript and supporting files. Source data files have been provided for Figures 5-10, 12, 14 and 15; the Supplement to Figure 8 and Tables 2-4.

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
