## [Decision Letter]

Thank you for submitting your article "SETD3 protein is the actin-specific histidine *N*-methyltransferase" for consideration by *eLife*. Your article has been reviewed by three peer reviewers, and the evaluation has been overseen by Philip Cole as the Senior Editor and Reviewing Editor. The following individual involved in review of your submission has agreed to reveal his identity: Paul R Thompson. Two other reviewers remain anonymous.

The reviewers have discussed the reviews with one another and the Reviewing Editor has drafted this decision to help you prepare a revised submission.

Summary:

The manuscript by Kwiatkowski et al. describes the discovery that SETD3 is the actin-specific histidine *N*-methyltransferase. The paper uses classical purification techniques and enzymatic assays to enrich for actin-specific histidine *N*-methyltransferase from rat muscle. Proteomic techniques then identified SETD3 as a candidate histidine *N*-methyltransferase, which was subsequently validated in vitro. The authors went on to develop CRISPR knockout lines and showed that this modification is important for maintaining cellular F-actin levels and for regulating glycolysis. Overall, this paper represents the beginnings of a field of study, and we expect that it will be widely appreciated. The authors should be roundly applauded for their comprehensive Materials and methods section as it demonstrates the rigor of the work.

Essential revisions:

1) Regarding Table 4, please provide kinetic data for the methylation of peptide mimics of histone H3 and actin in the table. If saturation is not achieved, provide estimates of k_cat_/K_M_ for each peptide. This is important for putting the catalytic efficiency in context with other enzymes that catalyze methyl transfer reactions.

2) The authors compared peptide substrates derived from actin and histone to compare SETD3 methylation activity on His and Lys. Since differences between actin protein and peptide are ~250-fold, it would be important to compare the histone H3 protein with actin protein as substrates for methyltransferase studies to put into better context the relative substrate recognition. In addition, it would be valuable to use docking to understand how SETD3 recognizes two different substrates for methyl transfer. Since the structure of human SETD3 is available, more efforts should be made to compare the structure with other well-characterized SET-domain lysine methyltransferases at the sequence and 3D levels. Given the different physical properties between Lys and His, the residues that favor Lys binding may not be conserved for His binding. More comparisons of this sort will strengthen this manuscript.

3) To access recombinant human/rat SETD3, the authors expressed these enzymes in mammalian cells, which also contain their endogenous histine methyltransferase activities. There is thus a small possibility that the histine methyltransferase activity comes from a polypeptide that can associate with SETD3 rather than intrinsic to SETD3 itself. To most effectively address this concern, the authors should access recombinant human/rat SETD3 with a single-point mutation in a conserved residue in the active site that is predicted to eliminate catalytic activity and demonstrate that such a mutation is sufficient to abolish the activity. The modeling effort described above (point 2) should allow for identification of such a mutant(s). Alternatively, SETD3 should be expressed in a cell strain which lacks endogenous His methyltransferase activity to authenticate that the enzymatic activity comes from SETD3.

4) To characterize the functional roles of SETD3-mediated actin His74me, only *Setd3*-deficient lines were examined. The limitation of this experiment is that it is possible that the full length protein of SETD3 rather than its catalytic activity accounts for the observed biological outcomes. Therefore, a recue experiment with active and catalytically dead SETD3 should be conducted to further validate the roles of SETD3. Alternatively, the authors could compare the functional outcomes of wild-type actin with those of His74X actin in wild-type and *Setd3*-deficient contexts.

5) To understand the scope of the SETD3 reaction, it would be valuable to incubate the enzyme with lysate and determine whether there are multiple substrates. If they could demonstrate that multiple proteins are methylated by this enzyme, it would greatly ramp up enthusiasm in the methyltransferase field for this PTM and enzyme function.

Other points:

6) Subsection “Characterization of Recombinant SETD3 Proteins”, last paragraph: The lack of methylation of yeast produced actin is a concern, however, the authors data are consistent with their conclusion that there must be a conformational difference that precludes methylation of this form. Does ATP inhibit methylation of refolded actin? Please comment on this possibility.

7) Ideally, the authors could develop an antibody to detect methylated histidine 73 in actin and then specifically test whether the SETD3 knockouts show less methylated actin. However, this experiment is not critical at this stage because their mass spectrometry is consistent with the loss of this modification in the knockout cells.

8) Introduction, second paragraph: Please provide information on Hpm1p homology to other known methyltransferases

9) Subsection “Generation of the HAP1 Cell Lines Deficient in SEDT3 by CRISPR/Cas9 Method”: The authors should specify whether the CRISPR knockouts represent single or double allele knockouts.

10) Subsection “Glucose Consumption and Lactate Production”: The authors use PCA for perchloric acid whereas earlier they used the molecular formula. Since PCA is used only once, HClO4 is preferred.

11) The authors should provide a figure depicting the overall reaction and should comment on what nitrogen gets methylated (if known). The side chain of His contains two nitrogen atoms. It is not clear whether one of the two or both nitrogen atoms can be methylated by SETD3. If the latter is the case, did the authors identify a di-methylated product?

12) Figure 1: The fraction numbers 200, 400, 900, 1400 are confusing and it is unclear why they are provided in this format instead of, for example, 1, 2, 3 etc.

13) Figure 4: Change the color of H73 in panel B as the current colors are not color blind friendly.

14) Figure 5 legend: The legend is incomplete and it is difficult to evaluate the data in the figure.

15) "Actin-specific histidine N-methyltransferase" and "specifically modifies H73 residue" are imprecise descriptions for the following two reasons: 1) Authors used actin as the only protein substrate for SETD3. It is unknown if SETD3 can methylate His residues in other proteins. 2) SETD3 is a known lysine methyltransferase for K4 and K36 of H3. Therefore, "dual methyltransferase" may be a better choice in this case.

16) There is about a 250-fold activity difference between peptide and protein substrate measured here. Such differences may be caused by different recognition efficiency and possibly a specificity shift to other methylation sites. The authors stated that the His73 residue was the only tri-deuterium-methylated peptide in the MS analysis (subsection “Characterization of Recombinant SETD3 Proteins”, third paragraph). It is not clear if mono- and di-deuterium-methylated sites have been scrutinized.

17) The authors applied 6 purification steps to isolate the active fraction for His methylation. It would be helpful to provide the rationale for each purification step. Also, it is not clear how the purification (-fold) in Table 2 was calculated.

18) Since only nucleotide-free actin is the substrate, is it possible that methylated His facilitates the binding of actin with nucleotide and affects the dynamics of polymerization and depolymerization?

19) Protein markers in some figures including Figure 5, 8, Figure 6—figure supplement 1 seem to be a drawing instead of an image.

20) Professional editing to improve the writing would be helpful.

21) To characterize that His73 is the only target site of SETD3 with MS with recombinant proteins, the authors mentioned that only a His73 peptide contains this modification. In this situation, the authors should describe the percentage of MS coverage of actin was achieved (hopefully close to 100%).

22) It would be appreciated if the authors could comment on the percentage of His73me-containing actin out of the total actin. In general, the higher the percentage is, the more likely it is that this methylation event is functionally important.

---

## [Author Response]

Essential revisions:1) Regarding Table 4, please provide kinetic data for the methylation of peptide mimics of histone H3 and actin in the table. If saturation is not achieved, provide estimates of k_cat_/K_M_ for each peptide. This is important for putting the catalytic efficiency in context with other enzymes that catalyze methyl transfer reactions.

We have now determined the kinetic parameters for the methylation of actin peptide H (an analogue of methylation sites in vertebrate actin) with both human and rat SETD3 and these data are now included into Table 4. However, we were unable to obtain reasonable results for the histone peptide H3N4, due to the extremely low activity of SETD3 enzymes towards peptide H3N4, resulting in a very week and highly non-reproducible radiolabeling of this peptide as well as a high background radioactivity (a consequence of [^3^H]SAM degradation).

This kind of problem was expected, since the activity of SETD3 on histone peptide H3N4 is at least 10-fold lower than it is on actin peptide H (cf. Figure 8 – —figure supplement 1) and close to 5000-fold lower than the SETD3 activity on actin protein (cf. Table 4). However we tried to overcome these limitations by using very high concentrations of histone peptide H3N4 (up to 40 mM) and increased the radio assay time, which resulted in a loss of the enzyme’s activity.

Together, these results further support our hypothesis that SETD3 is an actin-methylating enzyme rather than a histone methyltransferase in vivo, resembling, for example, SETD7 enzyme that displays limited activity on nucleosomes in vitro, and efficiently methylates numerous non-histone substrates in vivo [for review see Herz, Garruss and Shilatifard, 2013].

2) The authors compared peptide substrates derived from actin and histone to compare SETD3 methylation activity on His and Lys. Since differences between actin protein and peptide are ~250-fold, it would be important to compare the histone H3 protein with actin protein as substrates for methyltransferase studies to put into better context the relative substrate recognition. In addition, it would be valuable to use docking to understand how SETD3 recognizes two different substrates for methyl transfer. Since the structure of human SETD3 is available, more efforts should be made to compare the structure with other well-characterized SET-domain lysine methyltransferases at the sequence and 3D levels. Given the different physical properties between Lys and His, the residues that favor Lys binding may not be conserved for His binding. More comparisons of this sort will strengthen this manuscript.

We find the reviewers’ suggestions interesting and we thank them for it. However, we think it is unlikely that either (1) the comparison of SETD3 activity towards histone H3 and actin proteins or (2) the suggested docking experiments will be successful and help to elucidate the mechanism of substrate recognition by the methyltransferase studied. The reasons for this are stated below:

At this stage we still do not know what is the native form(s) of actin that is the in vivo substrate of SETD3. As we emphasize in the manuscript, native ATP-/ADP-actin, for which the crystallographic data is indeed available, is not a substrate for SETD3 in vitro. The enzyme methylates only non-physiological, quasi-native nucleotide-free actin of unknown structure, indicating that the nucleotide-free native actin, probably in complex with one or more as yet not identified stabilizing protein(s), is likely to be the actual substrate for SETD3 in vivo. The same is also true for histone H3 protein, since the purified histone H3 is not a substrate for SETD3 enzyme either, in agreement with the observation that the histone methyltransferase was shown to modify H3 protein only if it is incorporated into isolated native nucleosomes (Chen et al., 2013).

All these observations are in fact in good agreement with the current knowledge for the enzymes belonging to the family of SET domain-containing proteins. Many of these methyltransferases are found in complexes with numerous other proteins that are essential for their catalytic activity and substrate specificity. For example, to exert its catalytic activity, yeast histone H3 methyltransferase SET1 requires the presence of 7 different proteins that form a complex named COMPASS (Complex of Proteins Associated with Set1), and similar COMPASS-like complexes are formed by mammalian SET-domain containing methyltransferases [for review see Herz, Garruss and Shilatifard,2013].

To fulfill the suggestion from the reviewers diligently, the docking experiments should be performed employing 3D structures of SETD3 and the actin peptide H that could be reasonably predicted with, for example, PEP-Fold server. We have in fact started such experiments and ceased them eventually, when we identified a yet non-released PDB record by Horton et al. (PDB: 6MBL), describing a crystallographic structure of SETD3 with the actin peptide (but not with the actin protein!). Therefore, since this should provide the required results very soon, we have decided to drop these experiments.

We have however tried to strengthen the in vivo function of SETD3 as the actin-specific histidine *N*-methyltransferase, as suggested by the reviewers, by determining the methylation status of actin in tissues of SETD3 knock-out *Drosophila melanogaster* larvae, employing the tandem mass spectrometry. These results are now included in Table 5, showing that the knockout of SETD3 prevented methylation of all three different forms of actin (actin-42A, -57B and -87E) present in extracts from the fly larvae, while all H74-containg actin peptides (which corresponds to the H73 in the mammalian protein) derived from wild-type larvae were all methylated. We believe that these findings further confirm the molecular identification of SETD3 as the actin-specific histidine *N*-methyltransferase.

3) To access recombinant human/rat SETD3, the authors expressed these enzymes in mammalian cells, which also contain their endogenous histine methyltransferase activities. There is thus a small possibility that the histine methyltransferase activity comes from a polypeptide that can associate with SETD3 rather than intrinsic to SETD3 itself. To most effectively address this concern, the authors should access recombinant human/rat SETD3 with a single-point mutation in a conserved residue in the active site that is predicted to eliminate catalytic activity and demonstrate that such a mutation is sufficient to abolish the activity. The modeling effort described above (point 2) should allow for identification of such a mutant(s). Alternatively, SETD3 should be expressed in a cell strain which lacks endogenous His methyltransferase activity to authenticate that the enzymatic activity comes from SETD3.

We agree with the reviewers’ objections, and to provide an unambiguous answer, we have produced both rat and human recombinant SETD3 in *E. coli* and purified them with Ni-NTA affinity chromatography (see Figure 7—figure supplement 1.). As now shown in Figure 7, *E coli*-derived purified recombinant SETD3 enzymes methylate actin protein with specific activity similar to that determined for SETD3 expressed in mammalian COS-7 cells (≈ 5 nmol × min^-1^× mg^-1^ protein). We believe that, this together with the various knock out models solve the concerns of the reviewers that are stated above.

We have now included the activity of *E. coli*-produced SETD3 towards actin in the Results section (subsection “Characterization of Recombinant SETD3 Proteins”, first paragraph).

*4) To characterize the functional* roles *of SETD3-mediated actin His74me, only Setd3-deficient lines were examined. The limitation of this experiment is that it is possible that the full length protein of SETD3 rather than its catalytic activity accounts for the observed biological outcomes. Therefore, a recue experiment with active and catalytically dead SETD3 should be conducted to further validate the roles of SETD3. Alternatively, the authors could compare the functional outcomes of wild-type actin with those of His74X actin in wild-type and Setd3-deficient contexts.*

We agree that rescue experiments could help strengthen that the phenotypic observations in HAP1 cells are indeed due to a lack of SETD3 catalytic activity. We think however, that the facts mentioned below rather support a crucial role of the catalytic activity (instead of the suggested structural effects) for the observed biological outcomes of *Setd3-deficiency*.

First, proteomics data show that actin protein is at least 40-fold more abundant than SETD3 in mammalian cells (protein expression level: 1836,1 and 44,1 ppm for β-actin and SETD3, respectively, The MaxQuant DataBase), indicating that only catalytic amounts of SETD3 are present in cells. This becomes even more apparent when it is contrasted with corresponding data for the well-characterized actin-binding proteins. For example, human cofilin-1 and profilin-2, actin-interacting proteins that regulate the rate of its polymerization/depolymerization through “a structural mechanism” are present in cells at levels similar to that of actin (2859,7 and 1577,4 ppm for cofilin-1 and profilin-2, respectively, The MaxQuant DataBase), while tropomyosin β chain that binds to actin filaments and stabilizes them is present in about 20-fold excesses over the SETD3 in various human cells (804,0 ppm, The MaxQuant DataBase).

Secondly, Nyman and colleagues [Nyman et al., 2002] reported that purified actin filaments built of monomers lacking H73 methylation show a significantly increased rate of depolymerization. This effect is due to an inherent structural instability of non-methylated actin monomers (and not the loss of any actin-interacting protein), which the authors detected using a thermal stability assay. We have thus tested the thermostability of actin present in cell-free lysates of control (methylated actin) and SETD3-KO (non-methylated actin) HAP1 cells. This was performed by heating the lysates for 5 min and subsequent quantification of the still soluble (non-denatured) actin by western blotting. As now shown in a novel Figure 14, at temperatures between 30 and 46 °C, the methylated actin (WT) is clearly more stable than the protein devoid of such modification (SETD3-KO). These results unambiguously show that the loss of SETD3 enzymatic activity and not the protein itself is the cause of observed disturbances in the cytoskeleton integrity of SETD3 KO cells.

We have now included into the Results section a short paragraph describing the results of the thermostability assay (subsection “Absence of SETD3 reduces F-actin and increases glycolytic rate in SETD3-deficient HAP1 cells”, second paragraph).

Summing up, we believe that these data are a reasonable argument in support that the phenotype we have identified in *Setd3*-deficient human HAP-1 cells is indeed due to lack of catalytic action of SETD3, rather than a non-catalytic/actin stabilizing interaction of SETD3. However, we took into consideration the reviewer’s comment and have now introduced a short paragraph addressing these concerns in the Discussion section (subsection “Biological Importance of SETD3 Methyltransferase”).

5) To understand the scope of the SETD3 reaction, it would be valuable to incubate the enzyme with lysate and determine whether there are multiple substrates. If they could demonstrate that multiple proteins are methylated by this enzyme, it would greatly ramp up enthusiasm in the methyltransferase field for this PTM and enzyme function.

We agree with the reviewers’ suggestions, and such experiments have already been done and their results are freely available now. Briefly, Cohn and colleagues [Cohn et al., 2016] performed a proteomic screen using the ProtoArray platform to identify interactors of SETD3 among 9500 human proteins tested. They found 172 novel SETD3 interacting proteins (including actin!), belonging to very different ontology classes (nucleic acid binding, transcription factors, enzymes modulators, cytoskeleton protein, etc.). Next, the authors decided to concentrate on the transcription factor FoxM1, showing that SETD3 apparently methylates this protein, though the identification of methylated site was not performed.

In contrast to this, when we searched for other proteins (in addition to actin) that might have been differently methylated in their histidines among those identified by LCMS in the larvae extracts of the WT flies and SETD3 KO ones, we did not to see any others.

Together, these findings suggest that one should not exclude the possibility that SETD3 might have multiple substrates yet, but more dedicated work is necessary to provide an answer to this question which could be the subject of another investigation.

Other points:6) Subsection “Characterization of Recombinant SETD3 Proteins”, last paragraph: The lack of methylation of yeast produced actin is a concern, however, the authors data are consistent with their conclusion that there must be a conformational difference that precludes methylation of this form. Does ATP inhibit methylation of refolded actin? Please comment on this possibility.

In fact, we had already addressed that exact same question but had not included the results in the manuscript. We found that the addition of either 0.5 mM ATP or ADP (10-fold excess over the nucleotides concentration in the preparation of yeast produced actin) did not impact the activity of either human or rat SETD3 towards the refolded actin.

We have now introduced a short sentence into the Results section, that provides this information (subsection “Characterization of Recombinant SETD3 Proteins”, last paragraph).

7) Ideally, the authors could develop an antibody to detect methylated histidine 73 in actin and then specifically test whether the SETD3 knockouts show less methylated actin. However, this experiment is not critical at this stage because their mass spectrometry is consistent with the loss of this modification in the knockout cells.

We agree with the reviewers’ suggestion, and it is indeed an option that could replace the MS experiments, but we have not developed such antibody yet. However, we now show (by MS analysis) that the methylhistidine-containing actin is also absent from tissues of SETD3 knock-out *Drosophila melanogaster* larvae (see comment 2.).

8) Introduction, second paragraph: Please provide information on Hpm1p homology to other known methyltransferases

Hpm1p (also known as YIL110W) is the seven β-strand methyltransferase (7BS MTase), containing motifs I, post I and motif II that are hallmarks of this group of enzymes [Webb et al., 2010]. Recently Hpm1p has been classified in the so-called methyltransferase family 16 (MTF16), based on the presence of the conserved DXXY motif localized immediately downstream of Motif II [for review see Falnes et al., 2016]. The MTF16 group comprises mostly protein-lysine methyltransferases such as Mettl20 – 23 and Camkmt, with the noticeable exception of Hpm1p and Efm7 that methylate His243 of the yeast ribosomal protein Rpl3p ribosomal and the N-terminal glycine in the eukaryotic translation elongation factor 1 α (eEF1A), respectively. The MTF16 group appears to be distinct from both other known 7BS protein-lysine MTases (e.g., Dot1L, Mettl10, N6amt2) and non-protein methylating 7BS MTases (e.g. Mettl1, Comt, Gnmt). These observations illustrate well that in general, the 7BS MTase represent a diverse set of enzymes, both in terms of amino acid sequence and of protein structure. They all share the core 7BS fold, but the additional structural elements required to make up the catalytically active enzyme vary greatly.

We have now introduced a short paragraph into the Introduction section to provide information on the homology of Hpm1p to other members of methyltransferase family 16 (Introduction, second paragraph).

9) Subsection “Generation of the HAP1 Cell Lines Deficient in SEDT3 by CRISPR/Cas9 Method”: The authors should specify whether the CRISPR knockouts represent single or double allele knockouts.

The HAP1 cells have a haploid karyotype except for a disomy of chromosome 8 (25, XY, +8, PH^+^) and are therefore considered near-haploid. They are derived from the male chronic myelogenous leukemia (CML) cell line KBM-7 [Carette et al., Nature. 2011; 477:340-3]. Since *Setd3* gene is in chromosome 14, all generated *Setd3*-defficent HAP1 cells are single allele knockouts, showing the complete loss of SETD3 expression.

We have now clarified the text of Materials and methods section to highlight the near-haploid nature of HAP1 cells (subsection “Generation of the HAP1 Cell Lines Deficient in SEDT3 by CRISPR/Cas9 Method”, second paragraph).

10) Subsection “Glucose Consumption and Lactate Production”: The authors use PCA for perchloric acid whereas earlier they used the molecular formula. Since PCA is used only once, HClO4 is preferred.

We have changed the text accordingly.

11) The authors should provide a figure depicting the overall reaction and should comment on what nitrogen gets methylated (if known). The side chain of His contains two nitrogen atoms. It is not clear whether one of the two or both nitrogen atoms can be methylated by SETD3. If the latter is the case, did the authors identify a di-methylated product?

We detected only a mono-methylated histidine residue in both recombinant actin protein and synthetic peptide H by mass spectrometry analysis. To the best of our knowledge, no di-methylated histidine residues have ever been identified in proteins or metabolites.

We did not investigate which nitrogen of imidazole ring is methylated by SETD3 since this information is provided by Raghavan and collaborators [Raghavan, Lindberg and Schutt, 1992] who show that the actin-specific histidine methyltransferase catalyzes exclusively *N*τ-methylation of H73 residue.

We have now introduced a new Figure 1—figure supplement 2, describing the reaction catalyzed by the actin-specific histidine methyltransferase (SETD3), clearly showing the position of methylated nitrogen in the imidazole ring of histidine. The requested commentary is provided in the legend to this figure and is mentioned in the Discussion section (subsection “Molecular Identity of Rat Actin-specific Histidine N-Methyltransferase”, first paragraph).

12) Figure 1: The fraction numbers 200, 400, 900, 1400 are confusing and it is unclear why they are provided in this format instead of, for example, 1, 2, 3 etc.

We have now modified the figure and its legend accordingly.

13) Figure 4: Change the color of H73 in panel B as the current colors are not color blind friendly.

We have changed the color of H73 to black, one that is unambiguous both to colorblinds and non-colorblinds.

14) Figure 5 legend: The legend is incomplete and it is difficult to evaluate the data in the figure.

We have now modified both Figure 5 and its legend as requested by this reviewer.

15) "Actin-specific histidine N-methyltransferase" and "specifically modifies H73 residue" are imprecise descriptions for the following two reasons: 1) Authors used actin as the only protein substrate for SETD3. It is unknown if SETD3 can methylate His residues in other proteins. 2) SETD3 is a known lysine methyltransferase for K4 and K36 of H3. Therefore, "dual methyltransferase" may be a better choice in this case.

We understand the reviewers’ objection, but we would rather keep this description to indicate that: 1) SETD3 is the protein which indeed corresponds to the actin-specific histidine *N*-methyltransferase that is currently referenced as EC 2.1.1.85 and 2) actin is the only well-characterized substrate for SETD3, and much research effort is still needed to reliably identify other physiologically relevant substrates for this enzyme.

16) There is about a 250-fold activity difference between peptide and protein substrate measured here. Such differences may be caused by different recognition efficiency and possibly a specificity shift to other methylation sites. The authors stated that the His73 residue was the only tri-deuterium-methylated peptide in the MS analysis (subsection “Characterization of Recombinant SETD3 Proteins”, third paragraph). It is not clear if mono- and di-deuterium-methylated sites have been scrutinized.

We share the reviewers’ opinion that the difference in the activity of SETD3 towards peptide H and actin substrates is plausibly the consequence of the several thousand-fold lower affinity of the enzyme for the peptide substrate (see Table 4). Though, in our opinion, the simplest explanation of this phenomenon is that the more rigid H73-containg sensor loop of actin is much better fitted into the catalytic pocket of SETD3 than the unstructured peptide H.

We detected neither mono-deuterium (-CH_2_D, + 15.021926 Da) nor di-deuterium methylation (-CHD_2_, +16.028203 Da) of recombinant actin protein by Q-TOF mass spectrometry. This was in fact expected as the isotopic enrichment of utilized (S-methyl-d_3_)SAM (D-4093, C/D/N Isotopes Inc., Canada) was equal to 99% atom D.

17) The authors applied 6 purification steps to isolate the active fraction for His methylation. It would be helpful to provide the rationale for each purification step. Also, it is not clear how the purification (-fold) in Table 2 was calculated.

Since the molecular properties of the investigated methyltransferase were unknown, the choice of the purification strategy used is based both on our protein purification experience and on preliminary small scale trials. Once the purification conditions were optimized, the protocol was scaled-up for the FPLC as shown in the manuscript. The chromatographic methods used here, included ion-exchanger, gel filtration and affinity chromatography columns, which are those classically used when one is aiming to purify (and obtain a significantly enriched preparation) of a naturally abundant protein from a tissue.

The purification-fold is an enrichment in specific activity of SETD3 from one purification step to the next. So if the specific activity of SETD3 is 1 nmole/min/mg prot in the initial myofibrillar extract and 100 nmole/min/mg prot in the preparation obtained after 2 purification steps, the purification fold for SETD3 is 100/1 = 100-fold (there is a 100-fold enrichment in the activity of SETD3 after these two purification steps compared to the activity in the initial extract).

This way of measuring protein enrichment is classically used during protein purification experiments.

18) Since only nucleotide-free actin is the substrate, is it possible that methylated His facilitates the binding of actin with nucleotide and affects the dynamics of polymerization and depolymerization?

Nyman et al. [Nyman et al., 2002] showed that the rate constant for dissociation of ATP-H73A-actin is 10-fold higher than that for the ATP-actin, indicating that the methylated His73 may indeed facilitate/stabilize the binding of ATP to the actin monomer. However, based on the overall results, the authors concluded that the main biochemical role of methylated H73 is to delay the hydrolysis of ATP and P_i_-release, which prevents too rapid conversion of F-actin from ATP-state to ADP-state. Hence, an accelerated appearance of ADP-actin is likely the major cause of the observed disturbances in the integrity of nonmethylated F-actin.

We have now introduced a few sentences into Discussion section to clarify the importance of H73 methylation for the stability of ATP-actin (subsection “Biological Importance of SETD3 Methyltransferase”).

19) Protein markers in some figures including Figure 5, 8, Figure 6—figure supplement 1 seem to be a drawing instead of an image.

We thank the reviewers for this observation that in fact is correct. All western-blotting analyses were performed, employing chemiluminescence and signals acquisition with X-ray film, while the pattern of prestained protein ladder was copied from the blotting membrane onto the film using a set of felt-tip pens.

We have now modified the Materials and methods section to explain the presence of the hand-drawn protein ladder (subsection “Analytical Methods”).

20) Professional editing to improve the writing would be helpful.

The text has now been read and the English improved and edited by a senior scientist, fluent in English.

21) To characterize that His73 is the only target site of SETD3 with MS with recombinant proteins, the authors mentioned that only a His73 peptide contains this modification. In this situation, the authors should describe the percentage of MS coverage of actin was achieved (hopefully close to 100%).

The percentage of actin sequence coverage in MS analysis was equal to 75% (see Figure 9—source data 1). Though, this value is still quite far from 100%, taking into account that all known methyl-accepting sites in human actin (K18, K68, H73, K84, K191 and K326, see PhosphoSitePlus database) were covered and H73 was the only methylated residue in our analysis, and no radiolabeling of H73A actin was detected in the presence of SETD3, we think that our data fully support the conclusion that H73 is the only target site for SETD3 activity.

We took into consideration the reviewers’ comment and we have now modified the text of the Results section to provide required information (subsection “Characterization of Recombinant SETD3 Proteins”, third paragraph).

22) It would be appreciated if the authors could comment on the percentage of His73me-containing actin out of the total actin. In general, the higher the percentage is, the more likely it is that this methylation event is functionally important.

The data shown in the newly added Table 5 strongly indicates that 100% of actin is methylated in H73 in vivo, as non-methylated actin was not detected in extracts from *Setd3*-deficient fly larvae. However, to obtain a more comprehensive picture it would be easier to use an antibody that specifically recognizes meH73, which we have not developed, that could be used to quantify the amount of the methylated actin.

The functional importance of results shown in Table 5 has been now highlighted in the Results section (subsection “SETD3 is the β-actin H73 methyl transferase in HAP1 Human Cells and *Drosophila melanogaster*”, last paragraph.